# Generative Intervention Models for Causal Perturbation Modeling

**Nora Schneider** [1 2]  **Lars Lorch** [2]  **Niki Kilbertus** [1 3]  **Bernhard Schölkopf** [4 2 5]  **Andreas Krause** [2]

## Abstract

We consider the problem of predicting perturbation effects via causal models. In many applications, it is a priori unknown which mechanisms of a system are modified by an external perturbation, even though the features of the perturbation are available. For example, in genomics, some properties of a drug may be known, but not their causal effects on the regulatory pathways of cells. We propose a *generative intervention model (GIM)* that learns to map these perturbation features to distributions over atomic interventions in a jointly-estimated causal model. Contrary to prior approaches, this enables us to predict the distribution shifts of *unseen* perturbation features while gaining insights about their *mechanistic* effects in the underlying data-generating process. On synthetic data and scRNA-seq drug perturbation data, GIMs achieve robust out-of-distribution predictions on par with unstructured approaches, while effectively inferring the underlying perturbation mechanisms, often better than other causal inference methods.

## 1. Introduction

Decision-making and experiment design in the sciences require predicting how systems respond to perturbations. Since testing all perturbations in a design space is often intractable, expensive, or unethical, scientists need data-driven approaches for predicting the effects of unseen perturbations. For example, predicting how cells react to novel molecules can facilitate drug design in the vast space of candidate molecules (Sadybekov & Katritch, 2023; Tejada-Lapuerta et al., 2025). As in all learning tasks, however,

generalization to unseen instances requires an inductive bias (Ibragimov & Has'minskii, 1981). Perturbation effects are commonly predicted with either unstructured *black-box* models that predict the distribution shifts resulting from perturbations end-to-end or structured *mechanistic* models that characterize shifts as changes to an underlying causal data-generating process.

Unstructured approaches commonly treat the task of predicting perturbation effects as a statistical learning problem, given the perturbed samples and the features of the perturbation (e.g., Hetzel et al., 2022; Lotfollahi et al., 2023; Bunne et al., 2023). For instance, when predicting drug effects, they use features like dosage levels or molecular footprints as covariates to directly predict the perturbation response. This is practical because it allows making predictions for new perturbation features. However, these approaches do not explicitly model the underlying data-generating mechanisms and how they change under a perturbation (Schölkopf, 2022). As a consequence, unstructured models are challenging to interpret, difficult to use in experiment design, and impractical to use for perturbations with different feature spaces.

By contrast, mechanistic models characterize distribution shifts as interventions on a structured or causal model of the system (e.g., Pearl, 2009; Parascandolo et al., 2018; Gonzalez et al., 2024; Roohani et al., 2024). This perspective provides explanations for the predicted shifts as it explicitly infers the data-generating mechanisms and how they change under an intervention. Crucially, a causal model generalizes to new perturbations on the system—assuming we know the *atomic interventions* on the system components induced by a given perturbation. This assumption underlies most mechanistic approaches today and constrains them to settings where the perturbation targets are known (e.g., Maathuis et al., 2010) or, if unknown, inferred without considering the perturbation features (e.g., Mooij et al., 2020). By not leveraging the features of a perturbation as covariates, causal models currently cannot make predictions for novel, non-atomic perturbations.

In this work, we propose a causal modeling approach for predicting the effects of general perturbations, whose features may not directly reveal how the perturbation modifies the causal generative process. To achieve this, we train a generative intervention model (GIM) that maps the pertur-

---

[1]Technical University of Munich and Helmholtz Munich, Germany [2]Department of Computer Science, ETH Zurich, Switzerland [3]Munich Center for Machine Learning, Germany [4]MPI for Intelligent Systems, Tübingen, Germany [5]ELLIS Institute Tübingen, Germany. Correspondence to: Nora Schneider <nora.schneider@helmholtz-munich.de>.

*Proceedings of the 42$^{nd}$ International Conference on Machine Learning*, Vancouver, Canada. PMLR 267, 2025. Copyright 2025 by the author(s).

bation features to a distribution over atomic interventions in a jointly-learned causal model. The intervened causal generative process then models the distribution after perturbation. Since the GIM is shared across perturbations, it can systematically generalize to unseen perturbation features, similar to unstructured approaches, while providing mechanistic insights into how the data-generating process is altered. To evaluate the learned internal structure and perturbation predictions separately, our experiments evaluate GIMs in causal discovery, intervention target identification, and end-to-end perturbation effect prediction on synthetic data. Especially in nonlinear systems, GIMs outperform most causal inference approaches in structure and intervention target discovery, while predicting distribution shifts on par with unstructured approaches in out-of-distribution settings. On single-cell RNA sequencing (scRNA-seq) data of drug perturbations (Srivatsan et al., 2020), we show that GIMs can generalize effectively to new drug dosages and characterize the distribution shifts on a selection of genes quantitatively and qualitatively more accurately than the baselines.

## 2. Problem Statement and Related Work

We study a system of variables $\mathbf{x} \in \mathbb{R}^d$ with density $p(\mathbf{x})$ under perturbations of its generative process. A perturbation of $\mathbf{x}$ is represented by a feature vector $\boldsymbol{\gamma} \in \mathbb{R}^p$, which fully characterizes how $p(\mathbf{x})$ shifts under the perturbation; we denote the resulting perturbed density as $p(\mathbf{x}; \boldsymbol{\gamma})$. The perturbation features $\boldsymbol{\gamma}$ are observable and can, for instance, represent the chemical properties of a drug applied to cells or the context of an experiment. We observe data from $K$ experimental environments, each defined by a distinct perturbation applied to the system. For each environment $k$, we observe the corresponding perturbation features $\boldsymbol{\gamma}^{(k)}$ and the system's response, given by a data matrix $\mathbf{X}^{(k)}$ of i.i.d. samples from $p(\mathbf{x}; \boldsymbol{\gamma})$, e.g. the gene expression. Together, these observations form the full dataset $\mathcal{D} = \{(\mathbf{X}^{(k)}, \boldsymbol{\gamma}^{(k)})\}_{k=1}^{K}$. Given our dataset and an unseen perturbation $\boldsymbol{\gamma}^*$, we aim to model

$$p(\mathbf{x} \mid \mathcal{D}; \boldsymbol{\gamma}^*). \tag{1}$$

Our proposed framework leverages a causal model to characterize the (posterior predictive) perturbed density $p(\mathbf{x} \mid \mathcal{D}; \boldsymbol{\gamma})$ for arbitrary $\boldsymbol{\gamma}$. Before introducing our approach, we summarize relevant work on perturbation modeling and causal inference and discuss how they relate to our work.

**Autoencoders and optimal transport**  Common perturbation modeling approaches predict responses directly from $\boldsymbol{\gamma}$ without a mechanistic data-generating process. Autoencoders typically represent distribution shifts induced by perturbations in a learned latent space, some only applying to combinations of known perturbations (Lopez et al., 2018;

Lotfollahi et al., 2019), others generalizing to new perturbations by conditioning on $\boldsymbol{\gamma}$ (Yu & Welch, 2022; Hetzel et al., 2022; Lotfollahi et al., 2023). Optimal transport methods like CondOT (Bunne et al., 2022; 2023) learn transport maps and generalize by conditioning the map on $\boldsymbol{\gamma}$. Domain-specific foundation models (Cui et al., 2024; Hao et al., 2024) learn similar conditioning tokens. These approaches provide little insight into the system and perturbation mechanisms and are often highly specific to biochemistry.

**Causal models**  When using causal modeling to predict perturbation effects, causal inference methods typically focus on learning a structured, causal model $\mathcal{M}$ from data collected under different perturbations $\mathcal{D}$. Most methods assume that, for each perturbation, it is known how the system's causal mechanisms are modified. Under this assumption, $\mathcal{M}$ is estimated, e.g., through conditional independence tests (Spirtes et al., 2000) or model assumptions (Chickering, 2002; Zheng et al., 2018; Yang et al., 2018; Lorch et al., 2021). However, these methods cannot be applied when the underlying atomic modifications are unknown. Recent approaches aim at additionally inferring the atomic modifications that underlie the observations $\mathcal{D}$ (Mooij et al., 2020; Brouillard et al., 2020; Hägele et al., 2023; Squires et al., 2020). However, these methods infer the changes only for the perturbations in $\mathcal{D}$ and treat each perturbation independently, without leveraging the associated perturbation features $\boldsymbol{\gamma}$. As a result, they are unable to predict the effects of *unseen* perturbations $\boldsymbol{\gamma}^*$, leaving $p(\mathbf{x}; \boldsymbol{\gamma}^*)$ undefined for general perturbations. To do so, existing methods would require knowing the specific atomic interventions in $\mathcal{M}$ caused by $\boldsymbol{\gamma}^*$.

**Hybrid mechanistic models**  Some works combine mechanistic modeling with deep learning. However, these works either do not model perturbations (Parascandolo et al., 2018; Pervez et al., 2024) or are highly domain-specific, like CellOracle (Kamimoto et al., 2023), GEARS (Roohani et al., 2024) or PDGrapher (Gonzalez et al., 2024), who rely on expert knowledge to construct a graph neural network, which is not applicable in general settings.

## 3. Generative Intervention Models

In this section, we introduce generative intervention models, which form a novel class of densities $p(\mathbf{x}; \boldsymbol{\gamma})$ for perturbation modeling. Our approach leverages the inductive bias of a causal model, whose data-generating process characterizes the density of $\mathbf{x}$ under any perturbation $\boldsymbol{\gamma}$. To begin, we first describe causal models and how perturbations can be modeled as local modifications of the causal generative process. Then, we show how introducing an explicit generative model of these local modifications allows us to link the perturbation features $\boldsymbol{\gamma}$ to the causal model. This enables us to sample from $p(\mathbf{x}; \boldsymbol{\gamma}^*)$ for arbitrary perturbations $\boldsymbol{\gamma}^*$.

## 3.1. Causal Generative Process

To model the densities $p(\mathbf{x}; \boldsymbol{\gamma})$, we use a mechanistic model of the generative process of $\mathbf{x}$. Specifically, we assume that $\mathbf{x}$ is generated by a structural causal model (SCM; Pearl, 2009) with directed acyclic causal graph $\mathbf{G} \in \{0, 1\}^{d \times d}$ and parameters $\boldsymbol{\theta}$ that encode the causal relationships among the variables $\mathbf{x}$. The induced distribution $p(\mathbf{x}; \mathbf{G}, \boldsymbol{\theta})$ factorizes as a product of independent causal mechanisms

$$x_i \sim p_i(x_i \,|\, \mathbf{x}_{\mathbf{G}_i}; \boldsymbol{\theta}), \qquad (2)$$

with $p(\mathbf{x}; \mathbf{G}, \boldsymbol{\theta}) = \prod_{i=1}^{d} p(x_i \,|\, \mathbf{x}_{\mathbf{G}_i}; \boldsymbol{\theta})$. Each conditional distribution models a variable $x_i$ given its direct causes (parents) $\mathbf{x}_{\mathbf{G}_i}$. Under this generative process, each $x_i$ is independent of its nondescendants in $\mathbf{G}$ given its parents $\mathbf{x}_{\mathbf{G}_i}$.

Perturbations can be modeled as sparse mechanism shifts of the causal data-generating process, since perturbations typically only modify a few *causal* mechanisms of a system at once (Peters et al., 2017; Schölkopf, 2022). To capture this, we model perturbations of $\mathbf{x}$ as local (*atomic*) *interventions* on a sparse subset of the conditionals in Equation (2). An intervention that targets a variable $x_i$ replaces its causal mechanism with an interventional mechanism

$$x_i \sim \tilde{p}_i(x_i \,|\, \mathbf{x}_{\mathbf{G}_i}; \boldsymbol{\theta}, \boldsymbol{\psi}) \qquad (3)$$

with additional parameters $\boldsymbol{\psi}$, while not modifying any other conditionals of the factorization over $\mathbf{G}$. We may model various types of interventions, such as *hard* or *soft* interventions, which differ in how they modify the observational conditionals (Appendix A). We denote the variable indices targeted by an intervention by $\mathbf{I} \in \{0, 1\}^d$ and write

$$\mathcal{M} = \{\mathbf{G}, \boldsymbol{\theta}\}, \qquad \mathcal{I} = \{\mathbf{I}, \boldsymbol{\psi}\}.$$

For example, $\boldsymbol{\gamma}$ could represent the molecular features of a drug and $\mathcal{I}$ its targets and effects in the generative process of the gene expressions $\mathbf{x}$ of cells, which is characterized by $\mathcal{M}$. In summary, $p(\mathbf{x}; \mathcal{M})$ denotes the causal generative model and $p(\mathbf{x}; \mathcal{M}, \mathcal{I})$ the model perturbed by the atomic intervention $\mathcal{I}$. Next, we discuss how to bridge this approach to general perturbations $\boldsymbol{\gamma}$ and ultimately model $p(\mathbf{x}; \mathcal{M}, \boldsymbol{\gamma})$ for arbitrary features $\boldsymbol{\gamma}$.

## 3.2. A Generative Process of the Interventions

Perturbations of causal models can be modeled by atomic interventions $\mathcal{I}$. However, these atomic interventions are not observed in our problem setting, as we only observe the perturbation features $\boldsymbol{\gamma}$. Thus, to model the general perturbations $p(\mathbf{x}; \mathcal{M}, \boldsymbol{\gamma})$, the perturbation features $\boldsymbol{\gamma}$ have to be linked to an atomic intervention $\mathcal{I}$ such that, intuitively, $p(\mathbf{x}; \mathcal{M}, \mathcal{I}) = p(\mathbf{x}; \mathcal{M}, \boldsymbol{\gamma})$ for some $\mathcal{I}$ depending on $\boldsymbol{\gamma}$. Existing causal methods (e.g., Mooij et al., 2020; Hägele et al., 2023) directly infer $\mathcal{I}$ given samples from $p(\mathbf{x}; \boldsymbol{\gamma})$. In

particular, they independently infer a separate $\mathcal{I}$ for each perturbation in $\mathcal{D}$, without explicitly modeling how the features $\boldsymbol{\gamma}$ relate to the intervention $\mathcal{I}$. As a result, they cannot infer the intervention $\mathcal{I}$ for new features $\boldsymbol{\gamma}$ and thus cannot make predictions for the corresponding system response in practical prediction tasks.

Instead, we introduce an explicit generative process for the interventions $\mathcal{I}$ given the features $\boldsymbol{\gamma}$. This generative intervention model (GIM) is fully parameterized by $\boldsymbol{\phi}$ and shared across perturbations. It characterizes the distributions of the intervention targets and parameters conditioned on $\boldsymbol{\gamma}$ as

$$p(\mathcal{I}; \boldsymbol{\gamma}, \boldsymbol{\phi}) = p(\mathbf{I}; g_{\boldsymbol{\phi}}(\boldsymbol{\gamma})) p(\boldsymbol{\psi}; h_{\boldsymbol{\phi}}(\mathbf{I}, \boldsymbol{\gamma})), \qquad (4)$$

where $g_{\boldsymbol{\phi}}$ and $h_{\boldsymbol{\phi}}$ are arbitrary functions. Contrary to Hägele et al. (2023), we do not seek to directly infer $\mathcal{I}$ from $\mathcal{D}$. Instead, we view the interventions $\mathcal{I}$ as latent variables that contribute to the overall epistemic uncertainty given $\mathcal{D}$ and are marginalized out when modeling $p(\mathbf{x} \,|\, \mathcal{D}; \boldsymbol{\gamma})$. Specifically, GIMs model the generative process of $\mathbf{x}$ under a perturbation $\boldsymbol{\gamma}$ by marginalizing over its distribution of $\mathcal{I}$ as

$$p(\mathbf{x}; \mathcal{M}, \boldsymbol{\gamma}, \boldsymbol{\phi}) = \int p(\mathbf{x} \,|\, \mathcal{I}; \mathcal{M}) p(\mathcal{I}; \boldsymbol{\gamma}, \boldsymbol{\phi}) \, \mathrm{d}\mathcal{I} \qquad (5)$$

$$= \iint p(\mathbf{x} \,|\, \mathbf{I}, \boldsymbol{\psi}; \mathcal{M}) p(\mathbf{I}; g_{\boldsymbol{\phi}}(\boldsymbol{\gamma})) p(\boldsymbol{\psi}; h_{\boldsymbol{\phi}}(\mathbf{I}, \boldsymbol{\gamma})) \, \mathrm{d}\mathbf{I} \, \mathrm{d}\boldsymbol{\psi}.$$

Unlike previous causal inference approaches (Mooij et al., 2020; Hägele et al., 2023), GIMs can thus generalize to sampling from $p(\mathbf{x}; \boldsymbol{\gamma})$ for arbitrary unseen features $\boldsymbol{\gamma}$.

## 3.3. Posterior Predictive Distribution of GIMs

Given a data collection $\mathcal{D}$, our goal is to model the predictive density $p(\mathbf{x} \,|\, \mathcal{D}; \boldsymbol{\gamma})$ for any (unseen) perturbation $\boldsymbol{\gamma}$ (Section 2). From our generative model in Equation (5), we see that this requires posterior inference of the causal model $\mathcal{M}$ and GIM parameters $\boldsymbol{\phi}$, which map the perturbation features to the interventions in the causal model. For simplicity, we consider a crude approximation of the posterior, as indicated above, and simply learn maximum a posteriori (MAP) estimates of the causal model $\mathcal{M}$ and the GIM parameters $\boldsymbol{\phi}$:

$$p(\mathbf{x} \,|\, \mathcal{D}; \boldsymbol{\gamma}) = \int p(\mathbf{x} \,|\, \mathcal{M}, \boldsymbol{\phi}; \boldsymbol{\gamma}) p(\mathcal{M}, \boldsymbol{\phi} \,|\, \mathcal{D}) \, \mathrm{d}\mathcal{M} \, \mathrm{d}\boldsymbol{\phi}$$

$$\approx p(\mathbf{x} \,|\, \mathcal{M}^*, \boldsymbol{\phi}^*; \boldsymbol{\gamma}) \qquad (6)$$

with $\mathcal{M}^*, \boldsymbol{\phi}^* = \arg\max_{\mathcal{M}, \boldsymbol{\phi}} p(\mathcal{M}, \boldsymbol{\phi} \,|\, \mathcal{D})$. We note that, since $p(\mathcal{I} \,|\, \mathcal{D}, \boldsymbol{\gamma}) = \int p(\mathcal{I}; \boldsymbol{\gamma}, \boldsymbol{\phi}) p(\boldsymbol{\phi} \,|\, \mathcal{D}) \, \mathrm{d}\boldsymbol{\phi}$, posterior inference for GIMs enables performing *amortized* posterior inference of $\mathcal{I}$ given $\mathcal{D}$ downstream (cf. Hägele et al., 2023). In the next section, we show how to infer the MAP estimates $\mathcal{M}^*, \boldsymbol{\phi}^*$ with gradient-based optimization.

Unlike unstructured, black-box models, our approach infers the causal graph and mechanisms $\mathcal{M}$ as well as the

GIM parameters $\phi$ to approximate $p(\mathbf{x} \mid \mathcal{D}; \boldsymbol{\gamma})$. As a result, learning GIMs provides us with mechanistic insights into the generative process of the system, allows generalizing to new perturbations $\boldsymbol{\gamma}$, and even enables performing multiple perturbations $\boldsymbol{\gamma}$ jointly by generating their atomic interventions and applying them simultaneously in $\mathcal{M}$. In future work, GIMs may also quantify the uncertainty over its causal mechanisms (Lorch et al., 2021), e.g., to design the most informative next $\boldsymbol{\gamma}$; transfer its causal models to new applications that, e.g., perturb $\mathcal{M}$ with known atomic interventions $\mathcal{I}$ directly; and incorporate priors about the causal structure $\mathbf{G}$ to improve its predictions (e.g., Roohani et al., 2024). None of these are straightforward in black-box approaches (Section 2).

# 4. Inference

In this section, we describe how to jointly infer the MAP estimates of the causal model $\mathcal{M}$ and the GIM parameters $\phi$ with gradient-based optimization.

## 4.1. Model Components

Throughout this work, we make a set of modeling choices to render the joint MAP inference of $\mathcal{M}$ and $\phi$ tractable. We consider, e.g., linear Gaussian causal mechanisms given by

$$p_i(x_i \mid \mathbf{x}_{\mathbf{G}_i}; \boldsymbol{\theta}) = \mathcal{N}\big(x_i; \boldsymbol{\theta}_i^\top (\mathbf{G}_i \circ \mathbf{x}), \exp(\sigma_i)\big) \quad (7)$$

(e.g., Zheng et al., 2018) and nonlinear mechanisms with

$$p_i(x_i \mid \mathbf{x}_{\mathbf{G}_i}; \boldsymbol{\theta}) = \mathcal{N}\big(x_i; \mathrm{MLP}_{\boldsymbol{\theta}_i}(\mathbf{G}_i \circ \mathbf{x}), \exp(\sigma_i)\big) \quad (8)$$

(e.g., Brouillard et al., 2020; Lorch et al., 2021), where $\circ$ denotes elementwise multiplication and $\mathrm{MLP}_{\boldsymbol{\theta}_i}$ denotes a multi-layer perceptron (MLP) mapping $\mathbb{R}^d \rightarrow \mathbb{R}$ with trainable parameters $\boldsymbol{\theta}_i$. We learn the noise scale $\sigma_i$ for each variable jointly to mitigate data scale artifacts (Reisach et al., 2021; Ormaniec et al., 2025). This then fully specifies the causal generative model $p(\mathbf{x}; \mathcal{M})$ [1].

Similar to Brouillard et al. (2020) and Hägele et al. (2023), we model the targets $\mathbf{I}$ as independent Bernoullis and the parameters $\boldsymbol{\psi}$ as independent Gaussian distributions

$$p\big(\mathbf{I}; g_\phi(\boldsymbol{\gamma})\big) = \mathrm{Bern}\big(\mathbf{I}; \sigma\big(g_\phi(\boldsymbol{\gamma})\big)\big),$$
$$p\big(\boldsymbol{\psi}; h_\phi(\mathbf{I}, \boldsymbol{\gamma})\big) = \mathcal{N}\big(\boldsymbol{\psi}; h_\phi(\mathbf{I}, \boldsymbol{\gamma}), \eta_h^2\big).$$

The p.m.f. of Bern and p.d.f. of $\mathcal{N}$ are applied elementwise to each output of $g_\phi$ and $h_\phi$. The standard deviation $\eta_h$ is a hyperparameter, and $\sigma(\cdot)$ is the sigmoid function. In practice, $g_\phi$ and $h_\phi$ are MLPs. Together with $p(\mathbf{x}; \mathcal{M})$, we

---

[1] The specific choices in (7, 8) are not essential, as long as the dependence on $\mathbf{G}$ allows for a Gumbel-softmax relaxation. In our scRNA-seq experiments, we use zero-inflated $\log \mathcal{N}$ noise models.

can express the density of an observation $\mathbf{x}$ under model $\mathcal{M}$ and an intervention $\mathcal{I}$ as

$$p(\mathbf{x}; \mathcal{M}, \mathcal{I}) = \prod_{i=1}^{d} p_i(x_i \mid \mathbf{x}_{\mathbf{G}_i}; \boldsymbol{\theta})^{1-\mathbf{I}_i} \, \tilde{p}_i(x_i \mid \mathbf{x}_{\mathbf{G}_i}; \boldsymbol{\psi})^{\mathbf{I}_i}.$$

We describe two common choices for $\tilde{p}_i$ in Appendix A.

## 4.2. Maximum a Posteriori Estimation

With all causal model conditionals defined, we can compute the likelihood of the data $p(\mathbf{x}; \mathcal{M}, \boldsymbol{\gamma}, \phi)$ in Equation (5). Given the collection $\mathcal{D}$ of $K$ datasets $(\mathbf{X}^{(k)}, \boldsymbol{\gamma}^{(k)})$, our goal described in Section 3.3 is to infer the MAP estimate

$$\mathcal{M}^*, \phi^* = \arg\max_{\mathcal{M}, \phi} \log p(\mathcal{M}, \phi, \mathcal{D}).$$

For this, we define suitable priors over $\mathcal{M}$ and $\phi$ and then maximize the joint distribution over $\mathcal{D}$. We model $\mathcal{M}$ and $\phi$ as latent quantities, shared across the environments in $\mathcal{D}$, and the environments as conditionally independent given $\mathcal{M}$ and $\phi$ (Figure 1). Thus, the complete-data joint density is

$$\begin{aligned} \log p(\mathcal{M}, \phi, \mathcal{D}) = {} & \log p(\mathcal{M}) + \log p(\phi) \\ & + \sum_{k=1}^{K} \log p(\mathbf{X}^{(k)} \mid \mathcal{M}, \phi; \boldsymbol{\gamma}^{(k)}). \end{aligned} \quad (9)$$

Here, the log likelihood for $\mathbf{X}^{(k)}$ is obtained by summing the individual log likelihoods in Equation (5), since the samples are i.i.d. Before describing our priors and approximation of the likelihood, we first explain how we treat the discrete graph $\mathbf{G}$ in $\mathcal{M}$ during estimation.

**Differentiable inference of the causal model $\mathcal{M}$** We cannot perform gradient ascent on Equation (9) naively, because the graph matrix $\mathbf{G}$ in $\mathcal{M}$ is discrete and needs to be acyclic (Section 3). To address this, we use the continuous representation of $\mathcal{M}$ by Lorch et al. (2021). Specifically, we use a latent representation $\mathbf{Z} \in \mathbb{R}^{2 \times d \times p_Z}$ to model $\mathbf{G}$ probabilistically, where $p_Z$ is a parameter controlling the rank of the graphs that the distribution can represent. The graph distribution is defined as

$$p(\mathbf{G}) = \int p(\mathbf{Z}) p(\mathbf{G} \mid \mathbf{Z}) \, \mathrm{d}\mathbf{Z}$$

$$\text{with } p(\mathbf{G} \mid \mathbf{Z}) = \prod_{i=1}^{d} \prod_{j=1}^{d} \mathrm{Bern}\big(\mathbf{G}_{ij}; \sigma(\alpha \, \mathbf{z}_{0i}^\top \mathbf{z}_{1j})\big)$$

where $\alpha > 0$ is an inverse temperature parameter. Here, $\mathbf{z}_{0i}$ and $\mathbf{z}_{1j}$ denote the $i$-th and $j$-th rows of the first and second $(d \times p_Z)$ matrices in $\mathbf{Z}$, respectively. For $p_Z \geq d$, the conditional distribution $p(\mathbf{G} \mid \mathbf{Z})$ can represent any adjacency matrix without self-loops. We denote this continuous model

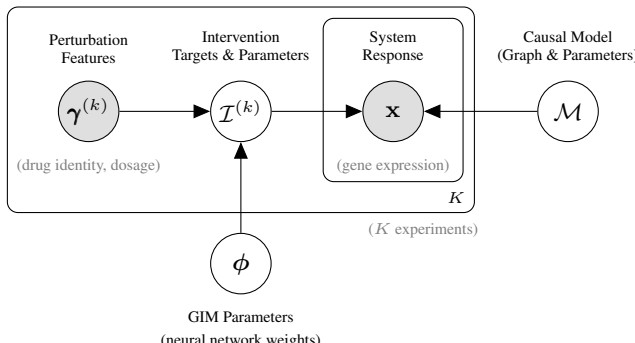

Figure 1. **Graphical model for the MAP estimation** of the causal model $\mathcal{M}$ and GIM parameters $\phi$. The atomic interventions $\mathcal{I}$ are marginalized out during inference. The dataset $\mathcal{D}$ consists of $K$ pairs of observed perturbation features $\boldsymbol{\gamma}^{(k)}$ and data matrices $\mathbf{X}^{(k)}$. For clarity, we depict $\boldsymbol{\gamma}^{(k)}$ as an observed random variable, but there is no prior over $\boldsymbol{\gamma}$, and we treat the perturbation features as a constant throughout. Gray labels indicate concrete examples of the variables in the context of drug perturbation experiments.

representation by $\mathcal{M}_\alpha = \{\mathbf{Z}, \boldsymbol{\theta}\}$. Under this generative model, the likelihood of the data $\mathcal{D}$ is given by

$$
\begin{aligned}
& p(\mathcal{D} \,|\, \mathcal{M}_\alpha, \boldsymbol{\phi}) \qquad\qquad \overbrace{\phantom{\mathbf{G}, \boldsymbol{\theta}}}^{\mathcal{M}} \qquad (10) \\
& = \int p(\mathbf{G} \,|\, \mathcal{M}_\alpha) \prod_{k=1}^K p(\mathbf{X}^{(k)} \,|\, \overbrace{\mathbf{G}, \boldsymbol{\theta}}, \boldsymbol{\phi}; \boldsymbol{\gamma}^{(k)}) \, \mathrm{d}\mathbf{G}
\end{aligned}
$$

(see Appendix B.1). The likelihood is the product of the GIM densities in Equation (5) over all observed perturbations in expectation over the graph given $\mathcal{M}_\alpha$. In this generative model, expectations over $p(\mathcal{M}_\alpha, \boldsymbol{\phi} \,|\, \mathcal{D})$ converge to those over $p(\mathcal{M}, \boldsymbol{\phi} \,|\, \mathcal{D})$ with $\mathbf{G} = \mathbb{1}[\mathbf{Z}_0 \mathbf{Z}_1^\top]$ as $\alpha \to \infty$ (Lorch et al., 2021). Hence, instead of $\log p(\mathcal{M}, \boldsymbol{\phi}, \mathcal{D})$, we optimize $\log p(\mathcal{M}_\alpha, \boldsymbol{\phi}, \mathcal{D})$ with respect to $\mathcal{M}_\alpha$.

**Priors** Our priors for learning GIMs capture the assumption that causal dependencies and interventions are sparse (Schölkopf, 2022). In addition to $L_2$ and sparsity regularization factors in all priors, the priors penalize cycles in $\mathbf{G}$. Specifically, we use the GIM parameter prior

$$
p(\boldsymbol{\phi}) \propto \mathcal{N}(\boldsymbol{\phi}; 0, \eta_{\boldsymbol{\phi}}^2) \prod_{k=1}^K \underbrace{\exp\big(-\mathbb{E}_{p(\mathcal{I} \,|\, \phi, \boldsymbol{\gamma}^{(k)})}\big[\beta_{\mathcal{I}} \|\mathbf{I}\|_1\big]\big)}_{\text{target sparsity}},
$$
(11)

which encourages sparse interventions via an $L_1$ penalty on the predicted targets. Additionally, we use the model prior

$$
\begin{aligned}
p(\mathcal{M}_\alpha) \propto\ & \mathcal{N}(\mathbf{Z}; 0, \eta_{\mathbf{Z}}^2) \mathcal{N}(\boldsymbol{\theta}; 0, \eta_{\boldsymbol{\theta}}^2) \mathcal{N}(\sigma; 0, \eta_\sigma^2) \\
& \cdot \underbrace{\exp\big(-\mathbb{E}_{p(\mathbf{G} \,|\, \mathbf{Z})}\big[\beta_{\mathcal{M}} \|\mathbf{G}\|_{1,1}\big]\big)}_{\text{mechanism sparsity}} \\
& \cdot \underbrace{\exp\big(-\mathbb{E}_{p(\mathbf{G} \,|\, \mathbf{Z})}\big[\lambda c(\mathbf{G}) + \tfrac{\mu}{2} c(\mathbf{G})^2\big]\big)}_{\text{acyclicity}}.
\end{aligned}
$$
(12)

All Gaussian densities are again applied elementwise, and $c(\mathbf{G})$ is the NO-BEARS acyclicity constraint (Lee et al., 2019), which is defined in Appendix D.4. The priors contain several hyperparameters concerning the sparsity ($\beta_{\mathcal{I}}$, $\beta_{\mathcal{M}}$), regularization ($\eta_{\boldsymbol{\phi}}$, $\eta_{\mathbf{Z}}$, $\eta_{\boldsymbol{\theta}}$, $\eta_\sigma$), which require tuning on held-out data but can often be shared.

**Stochastic optimization** Finally, to optimize the joint $\log p(\mathcal{M}_\alpha, \boldsymbol{\phi}, \mathcal{D})$ with respect to $\mathcal{M}_\alpha$ and $\boldsymbol{\phi}$, we approximate the integrals left implicit in the likelihood and priors. For this, we draw samples from both the graph model $p(\mathbf{G} \,|\, \mathcal{M}_\alpha)$ and the GIM $p(\mathcal{I}; \boldsymbol{\gamma}, \boldsymbol{\phi})$ to compute the log likelihood (10) and the log model prior (Equation (12)) via Monte Carlo integration. Overall, our MAP objective, the approximate continuous log joint density, is given by

$$
\begin{aligned}
\log p(\mathcal{M}_\alpha, \boldsymbol{\phi}, \mathcal{D}) \approx\ & \log p(\mathcal{M}_\alpha) + \log p(\boldsymbol{\phi}) \\
& + \log\bigg(\frac{1}{M} \sum_{m=1}^M \prod_{k=1}^K p(\mathbf{X}^{(k)} \,|\, \mathbf{G}^{(m)}, \boldsymbol{\theta}, \mathcal{I}^{(m,k)})\bigg),
\end{aligned}
$$

with $\mathbf{G}^{(m)} \sim p(\mathbf{G} \,|\, \mathcal{M}_\alpha)$ and $\mathcal{I}^{(m,k)} \sim p(\mathcal{I}^{(k)} \,|\, \boldsymbol{\phi}; \boldsymbol{\gamma}^{(k)})$ for $m \in \{1, \dots, M\}$. The log likelihood estimator is biased due to the logarithm but consistent as $M \to \infty$. To compute $p(\mathcal{M}_\alpha)$, we use the same approximation with independent samples from $p(\mathbf{G} \,|\, \mathcal{M}_\alpha)$. Since our causal generative process in Section 3.1 is only well-defined if $\mathbf{G}$ is acyclic, we follow Zheng et al. (2018) and maximize $\log p(\mathcal{M}_\alpha, \boldsymbol{\phi}, \mathcal{D})$ with respect to $\mathcal{M}_\alpha$ and $\boldsymbol{\phi}$ with the augmented Lagrangian method, which iteratively updates the acyclicity penalties $\lambda$ and $\mu$ (Appendix D.4). When $\mathcal{D}$ contains observational data with $p(\mathbf{x}; \boldsymbol{\gamma}) = p(\mathbf{x})$, we set $\mathbf{I} = 0$, which implies $p(\mathbf{x}; \mathcal{M}, \mathcal{I}) = p(\mathbf{x}; \mathcal{M})$. The MAP estimate for $\mathcal{M}$ is obtained via thresholding $\mathcal{M}_\alpha$ as $\mathbf{G} = \mathbb{1}[\mathbf{Z}_0 \mathbf{Z}_1^\top]$. Similarly, we use $\arg\max_{\mathcal{I}} p(\mathcal{I}; \boldsymbol{\gamma}, \boldsymbol{\phi})$ to sample from $p(\mathbf{x} \,|\, \mathcal{M}^*, \boldsymbol{\phi}^*; \boldsymbol{\gamma})$ at test time.

To compute the model gradients through the Monte Carlo samples during optimization, the Gaussians modeling $\boldsymbol{\theta}$ and $\boldsymbol{\psi}$ use the default reparameterization trick. The Bernoullis modeling $\mathbf{I}$ and $\mathbf{G}$ are reparameterized with the Gumbel-sigmoid (Maddison et al., 2017; Jang et al., 2017), except for the gradients with respect to $\boldsymbol{\theta}$, where we use the discrete $\mathbf{G}$. In Appendix B, we provide the explicit forms of the gradients of $\log p(\mathcal{M}_\alpha, \boldsymbol{\phi}, \mathcal{D})$ with respect to $\mathcal{M}_\alpha$ and $\boldsymbol{\phi}$.

**Identifiability** Under standard assumptions on the functional form and noise models, identifiability results for causal discovery with unknown interventions (Brouillard et al., 2020) apply directly to GIMs. Specifically, in the large-sample limit, the MAP estimates of GIMs recover the true intervention targets in the training data and a causal graph from the same interventional Markov equivalence class as the true graph (Yang et al., 2018). This result holds because, in the large-sample limit, the posterior is dominated by the likelihood, so the MAP estimates converge to

the same optimizers characterized in Brouillard et al. (2020). This further requires that the true interventional equivalence class contains a graph in the support of the prior over causal graphs. Moreover, if a mapping from perturbation features to intervention targets exists and is expressible by $\phi$, and the prior over $\phi$ supports this mapping, the true training targets can be recovered through optimization of $\phi$. Identifiability for unseen perturbations depends on the informativeness of the features $\gamma$, which we evaluate empirically Section 6.

## 5. Experimental Setup

We evaluate GIMs at predicting perturbation effects using both synthetic and real-world drug perturbation data.[2] GIMs estimate a causal model $\mathcal{M}$ of $\mathbf{x}$ and jointly learn to predict which interventions $\mathcal{I}$ in $\mathcal{M}$ characterize $p(\mathbf{x} \mid \mathcal{D}; \gamma)$. Although accurate estimates of these latent components are closely linked to the accuracy of $p(\mathbf{x} \mid \mathcal{D}; \gamma)$, their full identification is not strictly necessary for generalization. Thus, we first separately evaluate the inferred causal mechanisms and the predicted distributions on synthetic data with known generative processes. On scRNA-seq drug perturbations, where these mechanisms are unknown, we focus on evaluating the predicted distribution shifts.

### 5.1. Datasets

**Synthetic data**  To generate ground-truth systems, we create SCMs with additive Gaussian noise (Pearl, 2009). For this, we follow previous work (e.g., Brouillard et al., 2020) and sample DAGs from either an Erdős-Rényi (ER) (Erdős et al., 1960) or a scale-free (SF) (Barabási & Albert, 1999) distribution with $d = 20$ nodes and $2d$ edges in expectation. We then generate random linear or nonlinear mechanisms. Appendix D.1 describes the synthetic data in detail.

Generating synthetic perturbation mechanisms with features $\gamma$ that allow for generalization requires care. We draw inspiration from biological simulators (Dibaeinia & Sinha, 2020) and the biochemical perturbations evaluated later on and utilize randomly-generated *Hill functions* (Gesztelyi et al., 2012) to model different perturbations of the ground-truth SCM. Hill functions are nonnegative, saturating functions that satisfy $h(0) = 0$ and are used, e.g., to model transcription factor binding in gene regulation (Chu et al., 2009, see also Figure 2**C**). Inspired by drug compounds, each perturbation on the SCM consists of one or two random Hill functions, each assigned to a fixed but randomly-sampled target variable. The Hill functions define the shifts $\psi$ induced at their target variables $\mathbf{I}$ at a given *dosage* $c \in \mathbb{R}$. This enables us to evaluate algorithms at two levels of generalization. In the *partially out-of-distribution (OOD)* task, we test on perturbations with new dosages

$c \in \{0.25, 0.75, 1.25, 1.75, 2.25\}$ of the same intervention targets and Hill function mechanisms as in the training data. In the *fully OOD* task, we sample 20 new target and Hill function pairs distinct from the training set and perform perturbations at the training dosages $c \in \{0.5, 1, 1.5, 2\}$.

We create the features $\gamma$ by standardizing the concatenation of $\mathbf{I}$ and $\psi$, applying principal component analysis (PCA) across all training environments of a system, and keeping the top 15 principal components. This representation later also allows us to study the effect of the information content in $\gamma$ on generalization. The data for a given $\gamma$ is sampled from the interventional distribution of the SCM $p(\mathbf{x} \mid \mathbf{G}, \theta, \mathbf{I}, \psi)$.

**SciPlex3 drug perturbation data**  We also evaluate the predictive performance of GIMs on scRNA-seq data by Srivatsan et al. (2020). This dataset measures gene expressions under drug perturbations at four different dosages ($10nM$, $100nM$, $1\mu M$, and $10\mu M$) across three human cancer cell lines (A549, K562, MCF7). For our experiments, we focus on four drugs (Belinostat, Dacinostat, Givinostat, and Quinostat) from the epigenetic regulation pathway that were reported as among the most effective in this dataset and used in previous studies (Srivatsan et al., 2020; Hetzel et al., 2022). We apply standard single-cell preprocessing steps detailed in Appendix D.2. Given the high dimensionality of the data, we focus our analysis on the top 50 marker genes, i.e., genes with strongest post-perturbation effects overall. We construct the perturbation features $\gamma$ by combining the one-hot encodings of the drugs and dosages as well as the dosage values. For evaluation, we create test sets by holding out, one at a time, the highest dosage ($10\mu M$) of each drug, resulting in four unique training-test splits per cell type.

### 5.2. Baselines

We consider GIMs with different causal mechanisms. On the linear and nonlinear SCM data, GIMs learn Gaussian linear and MLP mechanisms as in Equations (7) and (8), respectively. On the scRNA-seq count data, we use zero-inflated log-normal MLP mechanisms (see Appendix D.4). We compare GIMs with black-box and causal inference approaches (Section 2). Our baselines consist of causal discovery methods for unknown interventions: BaCaDI (Hägele et al., 2023), GnIES (Gamella et al., 2022), UT-IGSP (Squires et al., 2020), and JCI-PC (Mooij et al., 2020; Spirtes et al., 2000). BaCaDI uses the same priors and likelihood as GIMs and a point estimate (BaCaDI*) and may be viewed as an ablation of GIMs that does not amortize the inference of $\mathcal{M}$ and $\mathcal{I}$. For JCI-PC, we evaluate the classical (JCI-PC) (Brouillard et al., 2020) and feature vector variant (JCI-PC Context), which extends the graph with context nodes for the perturbation features. Since all causal approaches except GIMs are limited to predictions for *seen* perturbations $\gamma$, we also benchmark two black-box approaches: CondOT

---

[2]Our code is available at https://github.com/NoraSchneider/gim

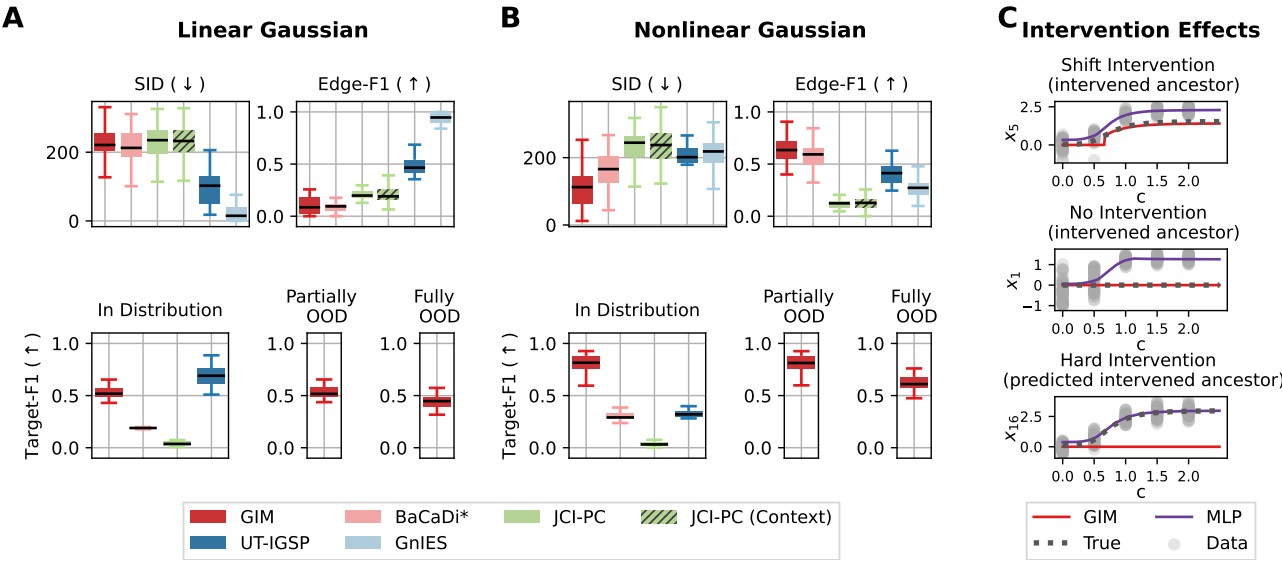

*Figure 2.* **Evaluating the learned causal structure and interventions.** **A** & **B**: For linear and nonlinear SCMs, SID and Edge-F1 scores of inferred causal graphs (top) and Target-F1 scores (bottom) for in- and out-of-distribution perturbation features $\boldsymbol{\gamma}$. GIMs outperform all baselines at inferring the causal structure and intervention targets in nonlinear systems, while also generalizing to OOD perturbation features. **C:** Example intervention effects predicted by GIMs and MLPs compared to the ground truth (Hill function) on nonlinear SCMs. Data plotted at $c$ correspond to the training data of $x_j$ at dosage $c$. Lines show the predicted (mean) intervention effects at all dosages $c \in [0, 2.5]$. In top and center panels, GIMs accurately infer the true intervention effects, even though the training data suggests differently (true interventions target ancestors). In bottom panel, the GIM misses an intervention but instead predicts an intervention on an ancestor.

([Bunne et al., 2022](#)) and an MLP that predicts the mean perturbation shifts directly from $\boldsymbol{\gamma}$. Appendices D.4 and D.5 provide details on GIMs and the baselines, respectively.

### 5.3. Metrics

To evaluate the predicted perturbed distributions $p(\mathbf{x}; \boldsymbol{\gamma}^*)$, we need to compare various methods, some without an analytical p.d.f. To enable this, we directly compare samples $\hat{\mathbf{X}}^*$ from the predicted distribution with true samples $\mathbf{X}^*$. We report their entropy-regularized Wasserstein distance ($W_2$), the Euclidean distance between their means (Mean distance), and the average negative log-likelihood of the true data under a kernel density estimate of the predicted data (KDE-NLL). For the scRNA-seq data, we also report the Pearson correlation of the mean differential expression after perturbation as commonly used in scRNA-seq analysis. For the synthetic data, we also evaluate the inferred causal mechanisms $\hat{\mathcal{M}}$ and interventions $\hat{\mathcal{I}}$. Given estimated and true graphs $\hat{\mathbf{G}}$ and $\mathbf{G}$, we compute the structural intervention distance (SID) ([Peters & Bühlmann, 2015](#)) and the F1 score of the edges (Edge-F1). Given predicted targets $\hat{\mathbf{I}}$, we report the F1 score given the true $\mathbf{I}$ (Target-F1) (see Appendix D.3).

### 6. Results

In Figures 2 and 3, we present the results on synthetic data with hard interventions. Figure 4 analyzes the effect of the information content in $\boldsymbol{\gamma}$, and Figure 5 reports the SciPlex3

results. Additional metrics and results for shift interventions and scale-free graphs are given in Appendix E.

**Explicitly modeling the perturbation mechanisms improves structure learning in nonlinear systems.** Figure 2 shows the accuracy of the estimated causal graphs compared to the ground-truth SCM that generated the data. For linear systems (**A**), GIMs, BaCaDI*, and both JCI-PC approaches predict similarly accurate graphs in terms of interventional implications (SID), while GnIES and UT-IGSP perform best. This aligns with previous findings that classical methods may outperform continuous optimization-based approaches on linear, normalized data, suggesting the gap to GIMs may result from factors discussed by [Reisach et al. (2021)](#) and [Ormaniec et al. (2025)](#). In nonlinear systems (**B**), GIMs significantly outperform the baselines across all metrics, including BaCaDI*, suggesting that the amortized inference model can enhance causal discovery. JCI-PC (Context) performs on par with JCI-PC, indicating that merely including perturbation features without modeling their actual mechanisms is not sufficient for enhancing causal discovery.

**GIMs generalize to unseen perturbation features and accurately infer their mechanisms of action.** Figure 2 also presents the Target-F1 scores for in-distribution, partially OOD, and fully OOD perturbations. GIMs significantly outperform all causal baselines in predicting intervention targets except for UT-IGSP, which performs best in linear settings. Moreover, the GIM framework is the only method

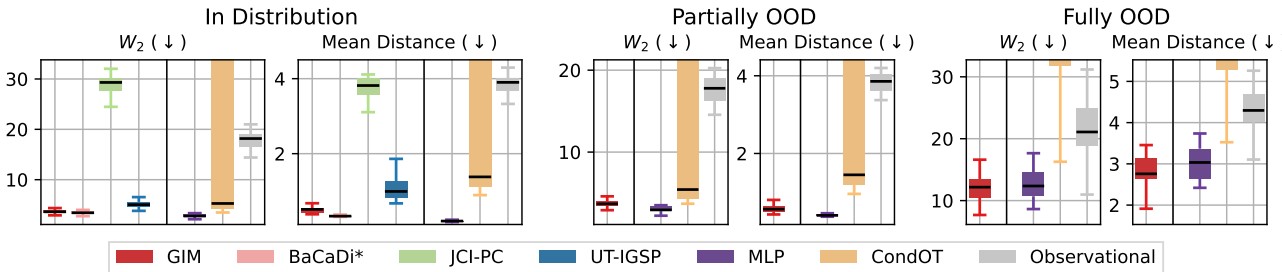

*Figure 3.* **Benchmarking the predicted distribution shifts on nonlinear Gaussian SCMs.** $W_2$ and Mean Distance for SCMs under perturbations with: seen dosages and targets (left, in distribution), novel dosages with seen targets (center, partially OOD), and novel targets with seen dosages (right, fully OOD). Metrics are medians over all perturbations for a given dataset. GIMs demonstrate robust predictive performance, matching the best causal baselines in distribution and performing on par with the best black-box approaches fully OOD Observational baseline uses the unperturbed distribution as a prediction. Vertical line separates mechanistic and black-box approaches. Boxplots show medians and interquartile ranges (IQR). Whiskers extend to farthest point within $1.5 \cdot$ IQR from the boxes.

capable of making predictions for unseen perturbations. For partially OOD perturbations, GIMs continue to identify true intervention targets with Target-F1 scores comparable to in-distribution settings. While a lower accuracy in the fully OOD setting is expected, GIMs achieve test-time scores that match or even exceed the baselines' in-distribution performance.

Figure 2 **C** illustrates the perturbation mechanisms learned by GIMs and the MLP baselines for selected variables alongside the true Hill functions defining the interventions. These examples emphasize the differences between black-box predictions and the GIM approach, which explicitly infers the mechanisms of action of a perturbation while allowing for end-to-end predictions. For instance, in the top and center panel, the samples of the four training perturbations suggest different variable shifts from those of the true underlying atomic intervention, because one of the ancestor variables is also intervened upon. While the MLP predicts shifts based on the observed data, the GIM correctly recovers the true Hill functions. In the bottom panel, GIM learns an incorrect intervention mechanism, yet it remains consistent with the estimated causal model. Although GIM misses an interven-

tion on a variable in this scenario, it correctly predicts an intervention on the parent of that variable in the estimated causal graph.

**GIMs predict distributions as accurately as black-box approaches fully OOD.** In Figure 3, we report the accuracies of the predicted distributions of each approach, irrespective of whether the underlying model $\mathcal{M}$ matches the ground truth SCM. For in-distribution perturbations, the end-to-end predictions of GIM are competitive with the best causal baselines across all evaluated metrics. All methods appear to capture meaningful perturbation effects, as their performances surpass both the observational prediction and CondOT. Despite not explicitly modeling the causal mechanisms, the MLP consistently performs best at predicting the effects of seen perturbations across all metrics.

Only GIMs and black-box methods can make predictions on unseen perturbations. Partially OOD, GIMs generalize well and achieve scores comparable to those of the in-distribution setting. On fully OOD perturbations, all baselines perform worse. Yet, the predictions of GIMs and the MLPs are better than the observational data and thus nontrivial, improving significantly over CondOT. While the MLP performs best in partially OOD, GIMs tend to yield better results for fully OOD interventions, while providing a mechanistic explanation for their predictions via their internal causal model.

**Generalization requires that the perturbation features $\gamma$ contain sufficient information.** To allow for generalization, the features $\gamma$ must allow for predictions about the (mechanistic) effects of unseen perturbations. To study this in GIMs, we leverage our proposed PCA representation to encode the atomic interventions as features $\gamma$ (Section 5.1). Figures 4 and 11 show how the predictive performance and target identification of GIMs degrades as we vary the number of PCA components contained in $\gamma$, keeping everything else fixed. When $\gamma$ contains sufficient information, GIMs effectively generalizes to completely unseen perturbations.

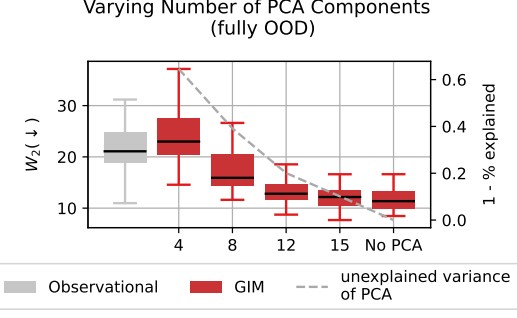

*Figure 4.* **Predictive accuracy ($W_2$) of GIMs relative to information contained in $\gamma$.** Fully OOD perturbations in nonlinear systems with hard atomic interventions. GIMs' predictive performance improves monotonically with the information content in $\gamma$.

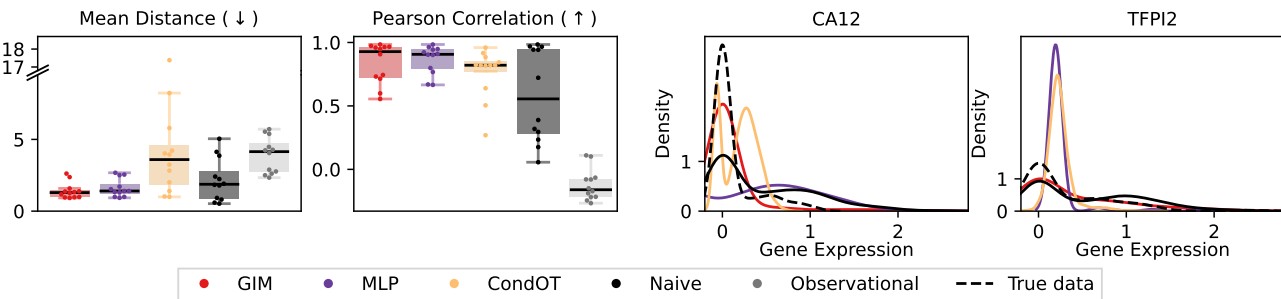

*Figure 5.* **Generalization to heldout drug-dosage combinations on the SciPlex3 scRNA-seq drug perturbation data (Srivatsan et al., 2020)**. Mean Distance and Pearson Correlation between predicted and true data (left). Examples of true and predicted marginal distributions on the 2 top-ranked marker genes for the Givinostat perturbation on A549 cells (right). *Naive* corresponds to predicting training data from the same drug and cell type at the closest lower dosage available in the training data ($1\mu M$). GIMs achieve the most robust predictions, quantitatively and qualitatively, and are the only mechanistic (causal) generative models applicable for unseen $\boldsymbol{\gamma}$.

**GIMs predict qualitatively accurate perturbation distributions in real-world settings.** In Figure 5, we present the results on the scRNA-seq perturbation data. The predictions of GIMs and the MLP baseline are most accurate as measured by Mean Distance and correlation (see Figure 5, left; for KDE-NLL, see Figure 12 in Appendix E.4). Here, we do not report $W_2$, as GIMs predictions are continuous p.d.f.s, whereas all other baselines' predictions (including MLP) as well as the true data are quasi-discrete (transformed and normalized integer counts), rendering distributional distances difficult to compare. Both GIMs and MLPs achieve better accuracy than a naive approach that predicts the response of the next-lower dosage in the training data. When inspecting the marginals for selected genes (Figure 5, right), we find that GIMs capture the distribution shifts qualitatively most accurately, leveraging the full (zero-inflated) generative model of the data. This shows that GIMs can provide meaningful predictions in settings with model mismatch.

## 7. Conclusion

We introduced generative intervention models, a general approach for using causal models to make predictions about perturbations of a data-generating process. GIMs learn to generate the atomic effects of a perturbation in a jointly-estimated causal model only based on its features. This allows us to tackle two key challenges in real-world decision-making: (1) inference of the causal mechanisms underlying a data-generating process, which makes GIMs inherently interpretable, and (2) generalization beyond the observed perturbations, where existing causal inference methods are not applicable. This dual capability distinguishes GIMs from existing work and enables experts to incorporate prior knowledge about the model and intervention mechanisms. Future work may use our approach in a fully Bayesian treatment (Lorch et al., 2021), to design optimally informative perturbations (Agrawal et al., 2019; Toth et al., 2022; Ailer et al., 2024), or scale it to larger systems (Lopez et al., 2022).

## Acknowledgements

This research was supported by the European Research Council (ERC) under the European Union's Horizon 2020 research and innovation program grant agreement no. 815943 and the Swiss National Science Foundation under NCCR Automation, grant agreement 51NF40 180545. We thank Scott Sussex for his valuable comments and feedback.

## Impact Statement

Our work studies causal models, which are relevant to any field aiming at predicting the effect of interventions, perturbations, or policy changes. If algorithms are grounded in the causal structure of the true data-generating process, they may be less prone to unexpected behavior or wrong conclusions, provided the assumptions underlying our approach are verifiable in the given application. Since our work aims at advancing machine learning and statistics, there are many potential societal and scientific consequences of our work, none which we feel must be specifically highlighted here.

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

# A. Additional Background

Atomic interventions in a causal model refer to any external manipulations or changes that modify the variables or causal mechanisms within the network (Peters et al., 2017; Vowels et al., 2022). Formally, an intervention on a variable $x_i$ corresponds to replacing the local conditional distribution $p_i$ in Equation (2) by a new interventional distribution $\tilde{p}_i$, while the conditionals of the variables not subject to the intervention remain unchanged (Schölkopf et al., 2021; Schölkopf, 2022). Thus, interventions are sparse or local in nature, meaning that they only affect the targeted variables and do not directly influence the other variables of a system.

We typically distinguish between two types of atomic interventions: *soft* and *hard* interventions (Peters et al., 2017; Pearl, 2009; Eberhardt et al., 2005). Soft (or imperfect) interventions modify the conditional probability distributions of the intervened-upon variable without changing the structure of the causal graph $\mathbf{G}$. This means that the functional dependencies on the parents remain. One example for soft interventions are so-called *shift* interventions (Rothenhäusler et al., 2015), which modify the conditional distributions of the target variables according to a shift vector $\boldsymbol{\psi} = (\boldsymbol{\psi}_1, ..., \boldsymbol{\psi}_d) \in \mathbb{R}^d$, so that

$$\tilde{p}_i(x_i = x + \boldsymbol{\psi}_i \,|\, \mathbf{x}_{\mathbf{G}_i}; \boldsymbol{\theta}, \boldsymbol{\psi}) = p_i(x_i = x \,|\, \mathbf{x}_{\mathbf{G}_i}; \boldsymbol{\theta}). \tag{13}$$

Hard (or perfect) interventions, on the other hand, eliminate the dependencies of the intervened variables on their causal parents so that the interventional conditional simplifies to

$$\tilde{p}_i(x_i | \mathbf{x}_{\mathbf{G}_i}; \boldsymbol{\theta}, \boldsymbol{\psi}) = \tilde{p}_i(x_i | \boldsymbol{\psi}), \tag{14}$$

ultimately altering the structure of the causal graph. Real world examples from biology include perfect gene knockout experiments for hard interventions and gene silencing for soft intervention (Squires et al., 2020).

# B. Derivations

## B.1. Continuous Model Representation

Similar to Lorch et al. (2021), we derive an expression for the joint distribution of the model using the latent probabilistic graph model. We get

$$\begin{aligned}
p(\mathcal{M}^\alpha, \boldsymbol{\phi}, \mathcal{D}) &= p(\mathbf{Z})p(\boldsymbol{\theta})p(\boldsymbol{\phi})p(\mathcal{D} \,|\, \mathbf{Z}, \boldsymbol{\theta}, \boldsymbol{\phi}) \\
&= p(\mathbf{Z})p(\boldsymbol{\theta})p(\boldsymbol{\phi}) \int p(\mathbf{G}, \mathcal{D} \,|\, \mathbf{Z}, \boldsymbol{\theta}, \boldsymbol{\phi}) \, \mathrm{d}\mathbf{G} \\
&= p(\mathbf{Z})p(\boldsymbol{\theta})p(\boldsymbol{\phi}) \int p(\mathbf{G} \,|\, \mathbf{Z}) \prod_{k=1}^K p(\mathbf{X}^{(k)} \,|\, \mathbf{G}, \boldsymbol{\theta}, \boldsymbol{\phi}; \boldsymbol{\gamma}^{(k)}) \, \mathrm{d}\mathbf{G} \\
&= p(\mathbf{Z})p(\boldsymbol{\theta})p(\boldsymbol{\phi}) \int p(\mathbf{G} \,|\, \mathbf{Z}) \prod_{k=1}^K \int p(\mathcal{I}^{(k)} \,|\, \boldsymbol{\phi}; \boldsymbol{\gamma}^{(k)}) p(\mathbf{X}^{(k)} \,|\, \mathbf{G}, \boldsymbol{\theta}, \mathcal{I}^{(k)}) \, \mathrm{d}\mathcal{I}^{(k)} \, \mathrm{d}\mathbf{G}
\end{aligned}$$

## B.2. Gradient Derivations

In the following, we detail the derivation of the gradients of the complete-data log joint density, $\log p(\mathcal{M}_\alpha, \boldsymbol{\phi}, \mathcal{D})$ (defined in Equation (9)), that allow us to use gradient-based optimization for the MAP estimation of the continuous representation of the causal model $\mathcal{M}_\alpha = \{\mathbf{Z}, \boldsymbol{\theta}\}$ and the parameters of the generative model $\boldsymbol{\phi}$.

**Derivation of $\nabla_{\mathbf{Z}} \log p(\mathcal{M}_\alpha, \boldsymbol{\phi}, \mathcal{D})$**

$$\nabla_{\mathbf{Z}} \log p(\mathcal{M}_\alpha, \boldsymbol{\phi}, \mathcal{D})$$

$$= \nabla_{\mathbf{Z}} \log p(\mathbf{Z}, \boldsymbol{\theta}) + \nabla_{\mathbf{Z}} \log p(\mathcal{D} \,|\, \mathbf{Z}, \boldsymbol{\theta}, \boldsymbol{\phi})$$

$$= \nabla_{\mathbf{Z}} \log p(\mathbf{Z}) + \nabla_{\mathbf{Z}} \log \left( \int p(\mathbf{G} \,|\, \mathbf{Z}) p(\boldsymbol{\theta} \,|\, \mathbf{G}) \, \mathrm{d}\mathbf{G} \right)$$

$$\quad + \nabla_{\mathbf{Z}} \log \left( \int p(\mathbf{G} \,|\, \mathbf{Z}) \prod_{k=1}^K \int p(\mathcal{I}^{(k)} \,|\, \boldsymbol{\phi}; \boldsymbol{\gamma}^{(k)}) p(\mathbf{X}^{(k)} \,|\, \mathbf{G}, \boldsymbol{\theta}, \mathcal{I}^{(k)}) \, \mathrm{d}\mathcal{I}^{(k)} \, \mathrm{d}\mathbf{G} \right)$$

$$= \nabla_{\mathbf{Z}} \log p(\mathbf{Z}) + \frac{\nabla_{\mathbf{Z}} \int p(\mathbf{G} \,|\, \mathbf{Z}) p(\boldsymbol{\theta} \,|\, \mathbf{G}) \, \mathrm{d}\mathbf{G}}{\int p(\mathbf{G} \,|\, \mathbf{Z}) p(\boldsymbol{\theta} \,|\, \mathbf{G}) \, \mathrm{d}\mathbf{G}} + \frac{\nabla_{\mathbf{Z}} \int p(\mathbf{G} \,|\, \mathbf{Z}) \prod_{k=1}^{K} \int p(\mathcal{I}^{(k)} \,|\, \boldsymbol{\phi}; \boldsymbol{\gamma}^{(k)}) p(\mathbf{X}^{(k)} \,|\, \mathbf{G}, \boldsymbol{\theta}, \mathcal{I}^{(k)}) \, \mathrm{d}\mathcal{I}^{(k)} \, \mathrm{d}\mathbf{G}}{\int p(\mathbf{G} \,|\, \mathbf{Z}) \prod_{k=1}^{K} \int p(\mathcal{I}^{(k)} \,|\, \boldsymbol{\phi}; \boldsymbol{\gamma}^{(k)}) p(\mathbf{X}^{(k)} \,|\, \mathbf{G}, \boldsymbol{\theta}, \mathcal{I}^{(k)}) \, \mathrm{d}\mathcal{I}^{(k)} \, \mathrm{d}\mathbf{G}}$$

During optimization, we sample i.i.d $\mathbf{G}^{(m)} \sim p(\mathbf{G} \,|\, \mathbf{Z})$, $\boldsymbol{\theta}^{(m)} \sim p(\boldsymbol{\theta} \,|\, \mathbf{Z})$ and $\mathcal{I}^{(m,k)} \sim p(\mathcal{I}^{(k)} \,|\, \boldsymbol{\phi}; \boldsymbol{\gamma}^{(k)})$ for $m \in \{1, ..., M\}$ to approximate the expectations. To ensure that we can compute the gradients through the samples, we use the reparameterization trick when sampling $\boldsymbol{\theta}$ and $\boldsymbol{\psi}$, and the Gumbel-sigmoid trick when sampling $\mathbf{G}$. This allows us to compute the gradients as follows:

$$\nabla_{\mathbf{Z}} \log p(\mathcal{M}_\alpha, \boldsymbol{\phi}, \mathcal{D})$$
$$\approx \nabla_{\mathbf{Z}} \log p(\mathbf{Z}) + \frac{\frac{1}{M} \sum_{m=1}^{M} \nabla_{\mathbf{Z}} p(\boldsymbol{\theta}^{(m)} \,|\, \mathbf{G}^{(m)})}{\frac{1}{M} \sum_{m=1}^{M} p(\boldsymbol{\theta}^{(m)} \,|\, \mathbf{G}^{(m)})} + \frac{\frac{1}{M} \sum_{m=1}^{M} \nabla_{\mathbf{Z}} \prod_{k=1}^{K} p(\mathbf{X}^{(k)} \,|\, \mathbf{G}^{(l)}, \boldsymbol{\theta}^{(m)}, \mathcal{I}^{(m,k)})}{\frac{1}{M} \sum_{m=1}^{M} \prod_{k=1}^{K} p(\mathbf{X}^{(k)} \,|\, \mathbf{G}^{(l)}, \boldsymbol{\theta}^{(m)}, \mathcal{I}^{(m,k)})}$$

**Derivation for $\nabla_{\boldsymbol{\theta}} \log p(\mathcal{M}_\alpha, \boldsymbol{\phi}, \mathcal{D})$**

$$\nabla_{\boldsymbol{\theta}} \log p(\mathcal{M}_\alpha, \boldsymbol{\phi}, \mathcal{D})$$
$$= \nabla_{\boldsymbol{\theta}} \log p(\mathcal{M}_\alpha) + \nabla_{\boldsymbol{\theta}} \log p(\mathcal{D} \,|\, \mathcal{M}_\alpha, \boldsymbol{\phi})$$
$$= \nabla_{\boldsymbol{\theta}} \log \left( \int p(\mathbf{G} \,|\, \mathbf{Z}) p(\boldsymbol{\theta} \,|\, \mathbf{G}) \, \mathrm{d}\mathbf{G} \right) + \nabla_{\boldsymbol{\theta}} \log \left( \int p(\mathbf{G} \,|\, \mathbf{Z}) \prod_{k=1}^{K} \int p(\mathcal{I}^{(k)} \,|\, \boldsymbol{\phi}; \boldsymbol{\gamma}^{(k)}) p(\mathbf{X}^{(k)} \,|\, \mathbf{G}, \boldsymbol{\theta}, \mathcal{I}^{(k)}) \, \mathrm{d}\mathcal{I}^{(k)} \, \mathrm{d}\mathbf{G} \right)$$

Similar to computing the gradients with respect to $\mathbf{Z}$, we sample from the respective distributions to approximate the expectations. We use the reparameterization trick when sampling $\boldsymbol{\theta}$ and $\boldsymbol{\psi}$.

**Derivation for $\nabla_{\boldsymbol{\phi}} \log p(\boldsymbol{\theta}, \mathbf{Z}, \boldsymbol{\phi} | \mathcal{D})$**

$$\nabla_{\boldsymbol{\phi}} \log p(\mathcal{M}_\alpha, \boldsymbol{\phi}, \mathcal{D}) = \nabla_{\boldsymbol{\phi}} \log p(\boldsymbol{\phi}) + \nabla_{\boldsymbol{\phi}} \log p(\mathcal{D} \,|\, \mathcal{M}_\alpha, \boldsymbol{\phi})$$
$$= \nabla_{\boldsymbol{\phi}} \log p(\boldsymbol{\phi}) + \nabla_{\boldsymbol{\phi}} \log \left( \int p(\mathbf{G} \,|\, \mathbf{Z}) \prod_{k=1}^{K} \int p(\mathcal{I}^{(k)} \,|\, \boldsymbol{\phi}; \boldsymbol{\gamma}^{(k)}) p(\mathbf{X}^{(k)} \,|\, \mathbf{G}, \boldsymbol{\theta}, \mathcal{I}^{(k)}) \, \mathrm{d}\mathcal{I}^{(k)} \, \mathrm{d}\mathbf{G} \right)$$

During optimization, we sample from the respective distributions to approximate the expectations. We use the reparameterization trick when sampling $\boldsymbol{\theta}$ and $\boldsymbol{\psi}$ and Gumbel-sigmoid trick when sampling $\mathbf{I}^{(k)}$.

## C. Computational Complexity

The computational complexity of GIMs is comparable to prior causal modeling approaches such as Hägele et al. (2023) and Brouillard et al. (2020) since we use similar causal model classes. The only additional overhead introduced by our new modeling framework is the forward pass through the GIM MLP, which does not affect the asymptotic complexity. Formally, the computational complexity of learning GIMs is:

$$\mathcal{O} \left( T \cdot \left[ d^2 \cdot (p_Z + H_{\mathcal{M}} + M \cdot n_{\text{power}}) + M \cdot n_{\text{total}} \cdot d \cdot L_{\mathcal{M}} \cdot H_{\mathcal{M}}^2 + K \cdot \left( L_{\text{GIM}} \cdot H_{\text{GIM}}^2 + H_{\text{GIM}} \cdot (p + d) + M \cdot d \right) \right] \right),$$

where:

- $T$: number of training steps,

- $d$: number of variables in $\mathbf{x}$,

- $p_Z$: rank parameter of $\mathbf{Z}$,

- $M$: number of MC samples,

- $n_{power}$: number of power iterations used in NO-BEARS acyclicity,

- $n$: total number of samples across environments

- $L_{GIM}, L_{\mathcal{M}}$: depth of GIM MLPs and causal mechanisms, respectively,

- $H_{GIM}, H_{\mathcal{M}}$: width of GIM MLPs and causal mechanisms, respectively,

- $p$: number of perturbation features in $\boldsymbol{\gamma}$,

- $K$: number of perturbation environments.

# D. Details on Experimental Setup

In the following section, we provide details on our synthetic data generation, preprocessing the Sciplex 3 dataset and implementation details on GIMs and all baseline approaches.

## D.1. Synthetic Data Generation

**Causal Mechanisms** For our synthetic data generation, we use linear and nonlinear Gaussian causal mechanisms as defined in Equation (7) and Equation (8), respectively. For linear causal mechanisms, linear functions model a variable's dependency on its parents and thus the parameters $\boldsymbol{\theta}$ represent these regression coefficients. For data generation, we randomly sample the parameters $\boldsymbol{\theta}$ uniformly from $[-3, -0.25] \cup [0.25, 3]$ to make sure they are bounded away from zero and exclude non-significant relationships. Further, we always use a constant variance for all variables $x_i$, thus $\exp(\sigma_i)^2 = \exp(\sigma)^2 = 0.1$. Similarly, for nonlinear causal mechanisms, a variable's dependency on its causal parents is defined via a nonlinear MLP. We thus have one MLP for each variable, so we have in total $d$ networks. We use MLPs with one hidden layer with 5 units and the $\tanh$ activation function. We sample the weights $\boldsymbol{\theta}$ from a standard Gaussian with mean 0 and variance 1. Again, we use $\exp(\sigma)^2 = 0.1$.

**Atomic Interventions** We generate random atomic interventions, by first sampling the intervention targets $\mathbf{I}$. Specifically, we uniformly sample the targets from all possible subsets of variabels containing exactly one or two elements. Next, we use randomly sampled Hill functions to compute the intervention parameters $\boldsymbol{\psi}$ acting on a fixed target variable. For each randomly-sampled target variable $x_i$, we separately sample $\lambda_i \sim \mathcal{U}[-6, -2] \cup [2, 6]$ and define its Hill function as $h_{\lambda_i}(c) = \lambda_i / (1 + (0.75/c)^4)$. Given the targets and their Hill functions, we then generate four distinct interventional environments for training by evaluating $h_{\lambda_i}(c)$ at the dosages $c \in \{0.5, 1, 1.5, 2\}$. Akin to drug perturbations, each of these four environments shares the same targets $\mathbf{I}$ but has different intervention effects $\boldsymbol{\psi}$, which are governed by the same Hill function evaluated at the different dosages $c$.

We consider two types of atomic interventions: hard and shift interventions, which are detailed in Appendix A. For settings with hard interventions, the intervention parameters $\boldsymbol{\psi}$, which are specified via the Hill functions, describe the mean of the interventional conditional distribution for a targeted variable. Specifically, the interventional conditional of a targeted variable is defined as $\tilde{p}_i(x_i|\boldsymbol{\psi}) = \mathcal{N}(x_i; \boldsymbol{\psi}, \exp(\sigma_{interv}))$. Contrary to the observational conditionals, we choose $\exp(\sigma_{interv})^2 = 0.5$ to ensure that the interventional distributions remain distinct from the observational ones even at small dosages, where the mean of the interventional conditional is close to zero, regardless of whether a variable has parents in the original graph. For settings with shift interventions, the intervention parameters $\boldsymbol{\psi}$, which are specified via the Hill functions, merely shift the targeted variables without introducing additional noise as defined in Equation (13) (Rothenhäusler et al., 2015).

**Synthetic Datasets** The training dataset consists of a total of 160 perturbational datasets (corresponding to 40 Hill functions considered at 4 different dosages), where each one has $n_k = 50$ samples, and one observational dataset with $n_0 = 800$ samples. The partially out-of-distribution test dataset consists of 200 perturbational contexts corresponding to the same 40 hill functions and intervention targets from the training dataset, but evaluated at 5 different dosages, $c \in \{0.25, 0.75, 1.25, 1.75, 2.25\}$. Finally, the fully out-of-distribution test dataset consists of 80 perturbational contexts corresponding to 20 newly sampled hill functions, which are evaluated at dosages $c \in \{0.5, 1, 1.5, 2\}$.

## D.2. SciPlex3 Data

Single-cell RNA sequencing data, like the SciPlex3 dataset (Srivatsan et al., 2020), usually comes with technical particularities that require careful preprocessing. Thus, we apply standard preprocessing methods. First, we normalize the data such that each cell has the same total count, equal to the median count of the unperturbed data. We then filter cells to retain only

those with at least 200 counts, and genes to keep only those expressed in at least 20 cells. Next, we apply a logarithmic transformation, $\tilde{x} = \log(1 + x)$. Since we focus on a subset of genes, we first identify the top 1000 highly variable genes, and subsequently reduce this set to the top 50 marker genes. To determine these, we rank all genes and select those with the highest absolute scores.

We limit our analysis to four drugs (Belinostat, Dacinostat, Givinostat, and Quinostat) from the epigenetic regulation pathway. For each drug perturbation, we construct corresponding perturbation features by concatenating the one-hot encoding of the drugs, one-hot encoding of the dosages, and rescaled dosage values, ensuring these values fall within $[0, 1]$. Specifically, we apply a hard-coded mapping of the dosages given by: $10nM \to 0.2$, $100nM \to 0.4$, $1\mu M \to 0.6$, and $10\mu \to 0.8M$. Thus, the perturbation vector is a 9-dimensional vector, with $\boldsymbol{\gamma} \in \{0,1\}^4 \times \{0,1\}^4 \times [0, 1]$.

In total, we consider 12 different experimental settings, corresponding to three cell types and four drugs. Specifically, in each setting, we hold out one drug at its highest dosage ($10\mu M$) as a test dataset to evaluate generalization. The remaining perturbations are used for training, resulting in a training dataset of $4 \times 3 + 3 + 1 = 16$ perturbations.

### D.3. Metrics

**Structural Interventional Distance (SID)**   We compare the predicted graph $\hat{\mathbf{G}}$ to the true causal graph $\mathbf{G}$ using the SID (Peters & Bühlmann, 2015), which quantifies their closeness based on their implied interventional distributions. It is defined as

$$SID(\hat{\mathbf{G}}, \mathbf{G}) := \#\left\{(i,j), i \neq j \;\middle|\; \begin{array}{ll} j \in \mathrm{DE}_i^{\hat{\mathbf{G}}} & \text{if } j \in \mathbf{x}_{\mathbf{G}_i} \\ \mathbf{x}_{\mathbf{G}_i} \text{ is not a valid adjustment set for } (i,j) \text{ in } \hat{\mathbf{G}} & \text{if } j \notin \mathbf{x}_{\mathbf{G}_i} \end{array}\right\},$$

where $\mathrm{DE}_i^{\hat{\mathbf{G}}}$ denotes the descendants of $i$ in $\hat{\mathbf{G}}$. We refer to Peters & Bühlmann (2015) and Peters et al. (2017) for a detailed definition of valid adjustment sets and the SID.

**F1-score of predicted edges (Edge-F1)**   Treating the inference problem as a classidication problem, we compute the F1-score of the edges in predicted graph $\hat{\mathbf{G}}$ with respect to the true causal graph $\mathbf{G}$. The F1-score is the harmonic mean of precision and recall, which is often prefered ove e.g. accuracy as the graphs are sparse. It is defined as

$$F1(\hat{\mathbf{G}}, \mathbf{G}) := \frac{2TP}{2TP + FP + FN}$$

where $TP$, $FP$ and $FN$ refer to the number of true positives, false positives and false negatives, respectively.

**F1-score of predicted intervention targets (Target-F1)**   We additionally report the F1-score of the predicted intervention targets with respect to the true intervention targets $F1(\hat{\mathbf{I}}, \mathbf{I})$, using the definition provided above.

**Entropy-regularized Wasserstein distance $W_2$**   We compare the predicted samples $\hat{\mathbf{X}}^*$ to the true samples $\mathbf{X}^*$ using the $W_2$ distance with a entropic regularization (Cuturi, 2013) provided in the OTT library (Cuturi et al., 2022) with

$$W_2(\mathbf{X}^*, \hat{\mathbf{X}}^*) := \left(\min_{\mathbf{P} \in U(\hat{\mathbf{X}}^*, \mathbf{X}^*)} \sum_{i=1}^{n} \sum_{j=1}^{m} \mathbf{P}_{ij} \|\mathbf{X}_i^* - \hat{\mathbf{X}}_j^*\|_2^2 - \epsilon H(\mathbf{P})\right)^{1/2}.$$

Here $n$ and $m$ are the number of samples in $\mathbf{X}^*$ and $\hat{\mathbf{X}}^*$, and $U$ is the set of ($n \times m$) transport matrices with $U = \{\mathbf{P} \in \mathbb{R}_{\geq 0}^{n \times m} : \mathbf{P}\mathbf{1}_m = 1/n \cdot \mathbf{1}_n \text{ and } \mathbf{P}^\top \mathbf{1}_n = 1/m \cdot \mathbf{1}_m\}$, with $\mathbf{1}_n$ denoting a $n$-dimensional vector of ones. $H(\mathbf{P})$ is the entropy with $H(\mathbf{P}) := -\sum_{ij} \mathbf{P}_{ij} \log \mathbf{P}_{ij} - 1$. $\epsilon$ is the regularization parameter and we always use $\epsilon = 0.1$.

**Mean Distance**   We compare the predicted samples $\hat{\mathbf{X}}^*$ to the true samples $\mathbf{X}^*$ using the Euclidean distance of their empirical means and is defined as

$$\text{Mean-Distance}(\hat{\mathbf{X}}^*, \mathbf{X}^*) := \|\hat{\mu}^* - \mu^*\|_2,$$

where $\hat{\mu}^*$ and $\mu^*$ are the empirical means of $\hat{\mathbf{X}}^*$ and $\mathbf{X}^*$, respectively.

**Average negative log-likelihood under kernel density estimation (KDE-NLL)** We use a kernel density estimation (KDE) with Gaussian kernel and bandwidth chosen according to Scott's rule (Scott, 2015) to obtain a density based on the predictions $\hat{\mathbf{X}}^*$. Then, we evaluate the average negative log-likelihood of the true samples $\mathbf{X}^*$ based on the KDE-estimate

$$\text{KDE-NLL}(\hat{\mathbf{X}}^*, \mathbf{X}^*) := -\frac{1}{n} \sum_{i=1}^{n} \log \hat{p}(\boldsymbol{X}_i^*)$$

$$\text{with } \hat{p}(\boldsymbol{X}_i^*) = \frac{1}{m \cdot bw} \sum_{j=1}^{m} K\left(\frac{\boldsymbol{X}_i^* - \hat{\boldsymbol{X}}_j^*}{bw}\right)$$

where $n$ and $m$ are the number of samples in $\mathbf{X}^*$ and $\hat{\mathbf{X}}^*$, respectively, $bw$ denotes the bandwidth and $K$ the Gaussian Kernel.

**Pearson Correlation between means** We compare the predicted samples $\hat{\mathbf{X}}^*$ to the true samples $\mathbf{X}^*$ using Pearson correlation on their means. Specifically, the Pearson correlation is defined as

$$\text{Pearson Correlation}(\hat{\mathbf{X}}^*, \mathbf{X}^*) := \frac{\sum_{i=1}^{d}(\hat{\mu}_i^* - \overline{\hat{\mu}^*})(\mu_i^* - \overline{\mu^*})}{\sqrt{\sum_{i=1}^{d}(\hat{\mu}_i^* - \overline{\hat{\mu}^*})^2 (\mu_i^* - \overline{\mu^*})^2}}.$$

Here we denote $\hat{\mu}^*$ and $\mu^*$ as the empirical means of $\hat{\mathbf{X}}^*$ and $\mathbf{X}^*$, respectively, and $\overline{\hat{\mu}^*}$ and $\overline{\mu^*}$ are the average values of $\hat{\mu}^*$ and $\mu^*$, respectively.

### D.4. GIM Implementation Details

In the following, we detail the implementation of GIMs used for our experiments reported in Section 6.

**Causal Mechanisms $\mathcal{M}$:** On synthetic data, we choose the causal inference model to have no mismatch with the generated data. Specifically, for linear data, we use the linear Gaussian inference model defined in Equation (7), and for nonlinear data, we use the nonlinear Gaussian inference model defined in Equation (8), with the MLPs containing a single hidden layer of 5 nodes and tanh-activation.

Because single-cell gene expression data is subject to dropout effects and remains positive even after our transformation, we model it with a zero-inflated log-normal ($\log \mathcal{N}$) inference model, which satisfies $\log(x) \sim \mathcal{N}$. The corresponding conditional for each variable $x_i \geq 0$ is given by

$$p(x_i \mid \mathbf{x}_{\mathbf{G}_i}; \boldsymbol{\theta}) = \pi_i \cdot \delta(x_i) + (1 - \pi_i) \log \mathcal{N}\big(x_i; \text{MLP}_{\boldsymbol{\theta}_i}(\mathbf{G}_i \circ \mathbf{x}), \exp(\sigma_i)\big), \tag{15}$$

where $\delta(x)$ is a Dirac delta at zero, and $\pi_i$ the learned probability of zero-inflation and $\sigma_i$ represents the learned noise scale for each variable $x_i$. Following the approach for nonlinear Gaussian mechanisms, we model the causal mechanisms with an MLP, selecting its architecture through hyperparameter tuning and using a tanh activation function.

In all experiments, we set the variance in the model priors in Equation (12) to $\eta_Z^2 = \frac{1}{d}$ (where $d$ is the number of system variables), $\eta_{\boldsymbol{\theta}}^2 = 0.1$, $\eta_{\sigma}^2 = 4$. In the prior of the GIM, we choose $\eta_{\phi}^2 = 0.1$.

In our proposed approach, we ensure the acyclicity of $\mathbf{G}$ via the model prior $p(\mathcal{M}_{\alpha})$ (see Equation (12)). Specifically, we use the continuous acyclicity constrained NO-BEARS proposed by Lee et al. (2019), who show that the spectral radius, i.e. the largest absolute eigenvalue, of an adjacency matrix is 0 if and only if the corresponding graph is acyclic. As suggested by Lee et al. (2019), we use power iteration to compute and differentiate through the largest absolute eigenvalue:

$$c(\mathbf{G}) := \rho(\mathbf{G}) \approx \frac{u^\top \mathbf{G} v}{u^\top v},$$

where we update for $t = 30$ steps according to the following update rule:

$$u \leftarrow u^\top \mathbf{G} / \|u^\top \mathbf{G}\|_2 v \leftarrow \mathbf{G} v / \|\mathbf{G} v\|_2,$$

and $u, v \in \mathbb{R}$ are initialized randomly.

**GIM**    We use the function $g_\phi$ to map the perturbation features to the probabilities of the Bernoulli distributions over the intervention targets $\mathbf{I}$. For this function, we consistently apply an MLP with two hidden layers, each containing 100 nodes, and a tanh activation function.

GIMs allows us to model different intervention types. On synthetic data, we model interventions in each setting according to the true underlying intervention type. For shift interventions, $\psi$ represents the shift parameters, and $h_\phi$ maps the perturbation features to the means of a Gaussian distribution over $\psi$. For hard interventions, $\psi$ includes both the interventional means and the interventional noise scale for a targeted variable. Here, $h_\phi$ consists of two separate MLPs: one mapping to the interventional means and the other to the interventional noise scale, both using the same architecture. For all synthetic settings, the MLPs for the intervention parameters $\psi$ have 2 hidden layers with 100 nodes and tanh activation function. On drug perturbation data, we assume hard atomic interventions. Thus, $\psi$ represents all parameters of the zero-inflated log-Gaussian conditional, that is the zero-inflation probability, the means and the noise scale. Thus $h_\phi$ consists of three separate MLPs, one for each parameter. We choose the architecture of the MLPs based on hyperparameter tuning and use a tanh-activation function.

**Optimization Details and Initialization**    We employ gradient-based optimization to obtain MAP estimates for $\mathcal{M}_\alpha$ and $\phi$. In Appendix B.2 we provided the gradients of the posterior allowing us to use Adam optimization (Kingma & Ba, 2015) with a learning rate of $0.001$. On synthetic data, we use 30000 steps and on drug perturbation data, we use 100000 steps. For the Monte Carlo approximations, we use a sample size of $n_{MC} = 128$. We apply a cosine annealing schedule to the coefficient, $\beta_{\mathcal{I}}$, which controls the sparsity of the intervention targets.

We initialize $\mathbf{Z}$ by sampling from a standard Gaussian distribution. When using linear causal mechanisms, we sample the initial parameters via $\boldsymbol{\theta}_{\text{init}} \sim \mathcal{N}(0, 0.1 \cdot \mathbf{I})$. For nonlinear mechanisms, we use the Glorot normal (Glorot & Bengio, 2010) to initialize the parameters $\boldsymbol{\theta}_{\text{init}}$. The log-variances are initialized with zero, $\boldsymbol{\theta}_i = \mathbf{0}$. Last, we use the Glorot initialization for the weights of GIM, $\phi$. This initialization procedure is important to avoid inducing a bias at the start of the optimization process.

We initialize the Lagrangian multiplier and the penalty coefficients with $\lambda_0 = 0$ and $\mu_0 = 1^{-9}$, respectively. Every 100 optimization steps, we evaluate based on 100 holdout samples if the objective converged. If it converges and the acyclicity constraint, $c(\mathbf{G})$ is not satisfied, we update the Lagrangian multiplier and penalty according to

$$\lambda_{t+1} = \lambda_t + \mu_t \cdot c(\mathbf{G}_t) \tag{16}$$

$$\mu_{t+1} = 2\mu_t. \tag{17}$$

### D.5. Baselines

**BaCaDi (Hägele et al., 2023):**    BaCaDi is a causal discovery approach for settings with unknown interventions and uses gradient-based variational inference. Thus it learns a posterior over the causal model and the intervention targets and parameters of each perturbation context. In order to obtain a point estimate, we implement only one particle in BaCaDi's inference procedure. Additionally, we adjust the inference model and its priors to match GIM's inference model. To distinguish our implementation from the original one, we call our implementation BaCaDi*. Through the modifications, BaCaDi* acts as a counterpart to GIM, which uses a similar inference model but does not model the perturbation mechanisms and thus cannot generalize to unseen perturbations.

**GnIES (Gamella et al., 2022):**    GnIES, an extension of the GES algorithm (Chickering, 2002), is a score-based causal discovery approach for settings with unknown interventions. GnIES optimizes a penalized likelihood score by alternately optimizing the set of intervention targets and the graph equivalence class. It assumes that the noise distribution of targeted variables changes across interventional contexts. As it assumes a linear model, there is a model mismatch for nonlinear systems. GnIES only estimates a graph equivalence class. Thus, for computing Edge-F1, we favor them by correctly orienting undirected edges when present in the ground truth graph. In case of a wrongly predicted edge, we orient it randomly. Since GnIES outputs a set of intervention targets for the entire dataset instead of perturbation-wise intervention targets, end-to-end perturbation predictions are not straightforward.

**JCI-PC (Mooij et al., 2020)**    The Joint Causal Inference (JCI) framework (Mooij et al., 2020) extends general causal discovery methods to accommodate data from various interventional contexts, including those with unknown interventions.

The authors suggest including information describing a system's context as additional nodes in the causal graph. Subsequently, standard causal discovery algorithms can be applied to the augmented graph. We implement this approach using the PC algorithm (Spirtes et al., 2000), which relies on a sequence of conditional independence tests to infer an equivalence class graph estimate. We use a Gaussian conditional independence test for all settings, since the KCI-test did not terminate in a reasonable time. We implement two types of JCI-PC: First, we add one node for each perturbational context (similar to the experiments conducted in (Hägele et al., 2023; Brouillard et al., 2020)). Second, we add the perturbation features as additional nodes to the causal graph (JCI-PC (Context)). Importantly, this approach does not infer the intervention targets for each perturbation context. As JCI-PC only estimates a graph equivalence class, we compute causal discovery metrics similar to GnIES. In order to evaluate the end-to-end perturbation predictions, we sample a graph from the equivalence class and use a linear Gaussian SCM with maximum likelihood parameter and variance estimates as the learned model (Hauser & Bühlmann, 2012).

**UT-IGSP (Squires et al., 2020):**    UT-IGSP extends the GSP (Yang et al., 2018; Wang et al., 2017) algorithm to settings with unknown intervention targets. Specifically, UT-IGSP learns a graph equivalence class and intervention targets by optimizing regularized local scores based on local conditional independence tests and permutation search. We compute causal discovery metrics similar to our GnIES implementation, as UTI-GSP also only estimates a graph equivalence class. To evaluate the end-to-end predictive performance, we follow the same procedure as for JCI-PC.

**CondOT (Bunne et al., 2022):**    CondOT learns a global optimal transport map conditioned on a perturbation's context variable, that effectively maps a measure onto another. This global map generalizes and ultimately allows us to make predictions for new contexts by conditioning on them. For our settings, we use the perturbation features $\gamma$ as the context variable and map the distribution of observational data $\mathcal{D}^{(0)}$ to the different perturbational ones $\mathcal{D}^{(k)}$. As the data of our experiment is already low-dimensional, we apply CondOT in the original dataspace.

**Multilayer Perceptron (MLP):**    We use an MLP to predict the shift of the perturbational distribution $\mathbf{X}^{(k)}$ relative to the observational one $\mathbf{X}_0$ from $\gamma_k$. Specifically, for each perturbational context, we calculate the shifts $s_k = \mu_k - \mu_0$, where $\mu_k$ and $\mu_0$ are the means of the datasets $\mathbf{X}^{(k)}$ and $\mathbf{X}^{(0)}$, respectively. The MLP is trained to predict these shifts from the perturbation features using a mean-squared error loss. When predicting the effects of new perturbations we shift the observational data $\mathbf{X}^{(0)}$ by the predicted shift, resulting in $\hat{\mathbf{X}}^* = \{x_i + s^*\}_{i=1}^{n_0}$.

### D.6. Hyperparameter Tuning

In our experiments on synthetic data, we benchmark GIM in different generative settings (linear/ nonlinear, hard/ shift interventions). For each setting, we perform a separate search of hyperparameters using 10 additionally generated datasets and choose the hyperparameters corresponding to the optional average KDE-NLL on the test datasets. Since the causal baselines are limited to predictions for seen interventions, we add the test dataset to the training data. Table 1 shows the range of hyperparameters investigated.

For our experiments on the Sciplex3 data, we use a similar approach. From each training dataset, we randomly hold out one drug at the extreme dosage and then select the hyperparameters to optimize the average KDE-NLL of the held-out samples. The search ranges are provided in Table 2

## E. Additional Experimental Results

### E.1. Additional results on Erdosrenyi Graphs

In Figure 6, we report an additional metric for measuring the accuracy of the predicted distribution, NLL-KDE, on nonlinear Gaussian SCMs with ER graphs, so the same settings reported already in Figure 3. Additionally, we show in Figure 6 the predictive accuracy for linear Gaussian SCMs.

In Figure 7, we report additional results for Gaussian SCMs with ER graphs, but with shift, atomic interventions underlying the perturbations. These results mostly match the results for hard interventions. We select hyperparameters as described above.

*Table 1.* **Hyperparameter tuning for the experiments on synthetic data in Section 6.**

| Method | Hyperparameter | Search Range |
|---|---|---|
| GIM | Sparsity graph | $\beta_{\mathcal{M}} \in \{50, 100, 500\}$ |
| | Sparsity intervention targets | $\beta_{\mathcal{I}} \in \{1, 10, 50, 100\}$ |
| | Inverse temperature sigmoid | $\tau \in \{0.01, 0.1, 1, 10\}$ |
| JCI-PC | Significance level | $\alpha \in \{1e-7, 1e-6, 1e-5, 1e-4, 0.001, 0.01, 0.1\}$ |
| JCI-PC (Context) | Significance level | $\alpha \in \{1e-7, 1e-6, 1e-5, 1e-4, 0.001, 0.01, 0.1\}$ |
| GnIES | Penalization parameter | $\lambda \in \{\text{BIC penalization}, 0.5, 1, 1.5, 2, 2.5, 3, 4, 6, 8\}$ |
| UT-IGSP | Significance level (CI) | $\alpha_{CI} \in \{1e-6, 1e-5, 1e-4, 0.001, 0.01, 0.1\}$ |
| | Significance level (invariance) | $\alpha_{inv} \in \{1e-6, 1e-5, 1e-4, 0.001, 0.01, 0.1\}$ |
| BaCaDi | Sparsity graph | $\beta_{\mathcal{M}} \in \{50, 100, 500\}$ |
| | Sparsity intervention targets | $\beta_{\mathcal{I}} \in \{1, 10, 50, 100\}$ |
| | Inverse temperature sigmoid | $\tau_G \in \{0.01, 0.1, 1, 10\}$ |
| MLP | Hidden layers | $n \in \{[50], [50, 50], [100], [100, 100], [100, 100, 100]\}$ |
| | Learning rate | $l \in \{0.001, 0.01\}$ |
| | Number of iterations | $n_{iter} \in \{1000, 5000, 10000, 20000, 40000\}$ |
| CondOT | Hidden layers | $n \in \{[64], [64, 64], [128, 128], [64, 64, 64, 64]\}$ |
| | Learning rate | $lr \in \{0.001, 0.01\}$ |
| | Number of iterations | $n_{iter} \in \{60000, 80000, 100000\}$ |

*Table 2.* **Hyperparameter tuning for the experiments on SciPlex3 data in Section 6.**

| Method | Hyperparameter | Search Range |
|---|---|---|
| GIM | Inference Model: Hidden layers | $n_{inf} \in \{[20], [20, 20], [20, 20, 20]\}$ |
| | GIM: Hidden layers | $n_{GIM} \in \{[100], [100, 100], [100, 100, 100]\}$ |
| | Sparsity graph | $\beta_{\mathcal{M}} \in \{0.1, 1, 10\}$ |
| | Sparsity intervention targets | $\beta_{\mathcal{I}} \in \{0.1, 1, 10\}$ |
| MLP | Hidden layers | $n \in \{[50], [50, 50], [100], [100, 100], [100, 100, 100]\}$ |
| | Learning rate | $l \in \{0.001, 0.01\}$ |
| | Number of iterations | $n_{iter} = \{1000, 5000, 10000, 20000, 40000\}$ |
| CondOT | Hidden layers | $n \in \{[64], [64, 64], [128, 128], [64, 64, 64, 64]\}$ |
| | Learning rate | $lr \in \{0.001, 0.01\}$ |
| | Number of iterations | $n_{iter} \in \{60000, 80000, 100000\}$ |

### E.2. Scale-Free Graphs

In Figure 8 and Figure 9 we report experimental results for synthetic data with scale-free graphs using hard atomic and shift atomic interventions, respectively. The experimental setup is the same as described in Section 5 with the only difference being that we use the hyperparameters that were optimized for settings with ER graphs for GIM and all other baselines. The results are mostly consistent with the ones on ER graphs and align with findings discussed in Section 6.

### E.3. Impact of Hyperparameter Selection and Feature Information on Model Performance

As discussed in Appendix D.6, we select the hyperparameters to optimize the predictive accuracy of the perturbed distribution, as measured by KDE-NLL. The accuracy of the estimated causal structure $\mathcal{M}$ is expected to be closely related to the predictive accuracy. Yet, full identification is not strictly necessary for good predictions of the perturbed density under $\gamma$. To put it in perspective, we report in Figure 10 the causal discovery metrics for GIM and all causal baselines, when selecting the hyperparameters based on Edge-F1. GIM and all causal baselines achieve higher accuracy compared to the setting where hyperparameters are selected based on KDE-NLL. However, in many real-world settings, the true causal model is unknown

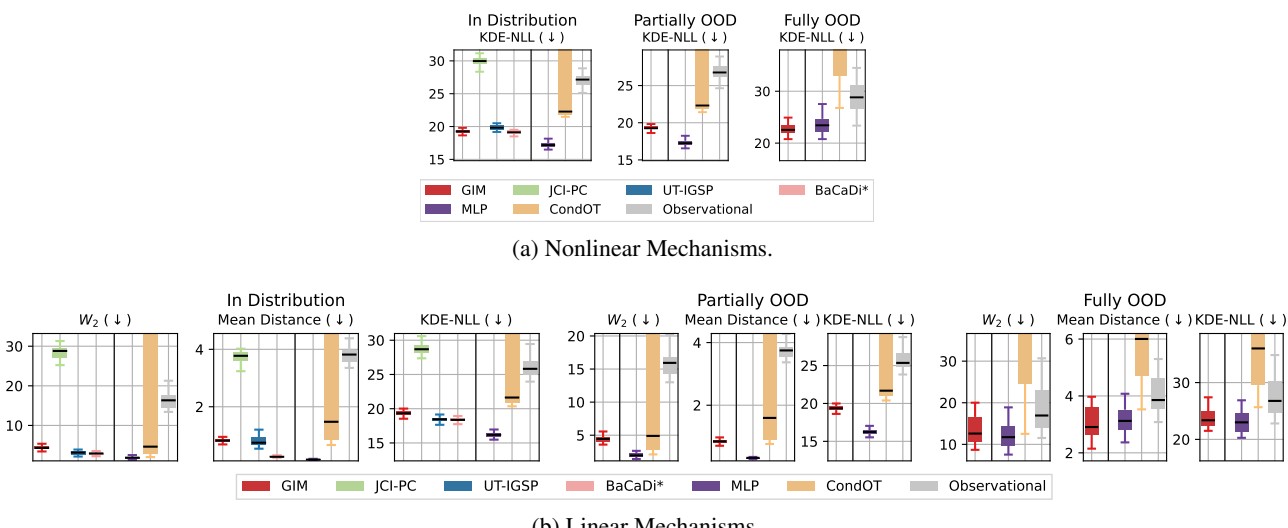

(a) Nonlinear Mechanisms.

(b) Linear Mechanisms

*Figure 6.* **Benchmarking the predicted distribution shifts on Gaussian SCMs with ER graphs and hard interventions.** (a) KDE-NLL for SCMs with nonlinear mechanisms. (b) $W_2$, Mean Distance and KDE-NLL for SCMs with linear mechanisms. We show perturbations with training dosages and targets (left, in distribution), novel dosages with seen targets (center, partially OOD), and novel targets with seen dosages (right, fully OOD). Metrics are medians over all perturbations for a given dataset. Observational baseline uses the unperturbed distribution as a prediction. Vertical line separates mechanistic and black-box approaches. Boxplots show medians and interquartile ranges (IQR). Whiskers extend to the farthest data point within $1.5 \cdot \text{IQR}$ from the boxes.

and thus the Edge-F1 is not accessible for selecting the hyperparameters. Further, the goal of our work is to use a causal model to ultimately predict the perturbed distribution and thus choosing hyperparameters based on metrics measuring the predictive accuracy might be more suited.

In Figure 11 we additionally report the accuracy of the inferred intervention targets when varying the number of principal components used for constructing the perturbation features. While for in-distribution perturbations, the performance of GIM converges with about 8 principal components, more information in is required for generalizing.

### E.4. Additional results for SciPlex3 (Srivatsan et al., 2020)

In Figure 12, we report the KDE-NLL for the predicted distribution of GIM and baseline approaches on SciPlex 3 data. Additionally, in Figure 13, we show the marginal histograms of the GIM and MLP prediction in one selected perturbation scenario. This visual example shows how the MLP merely shifts the observational distribution, while GIM, using the zero-inflated model, qualitatively better matches the true perturbations distributions.

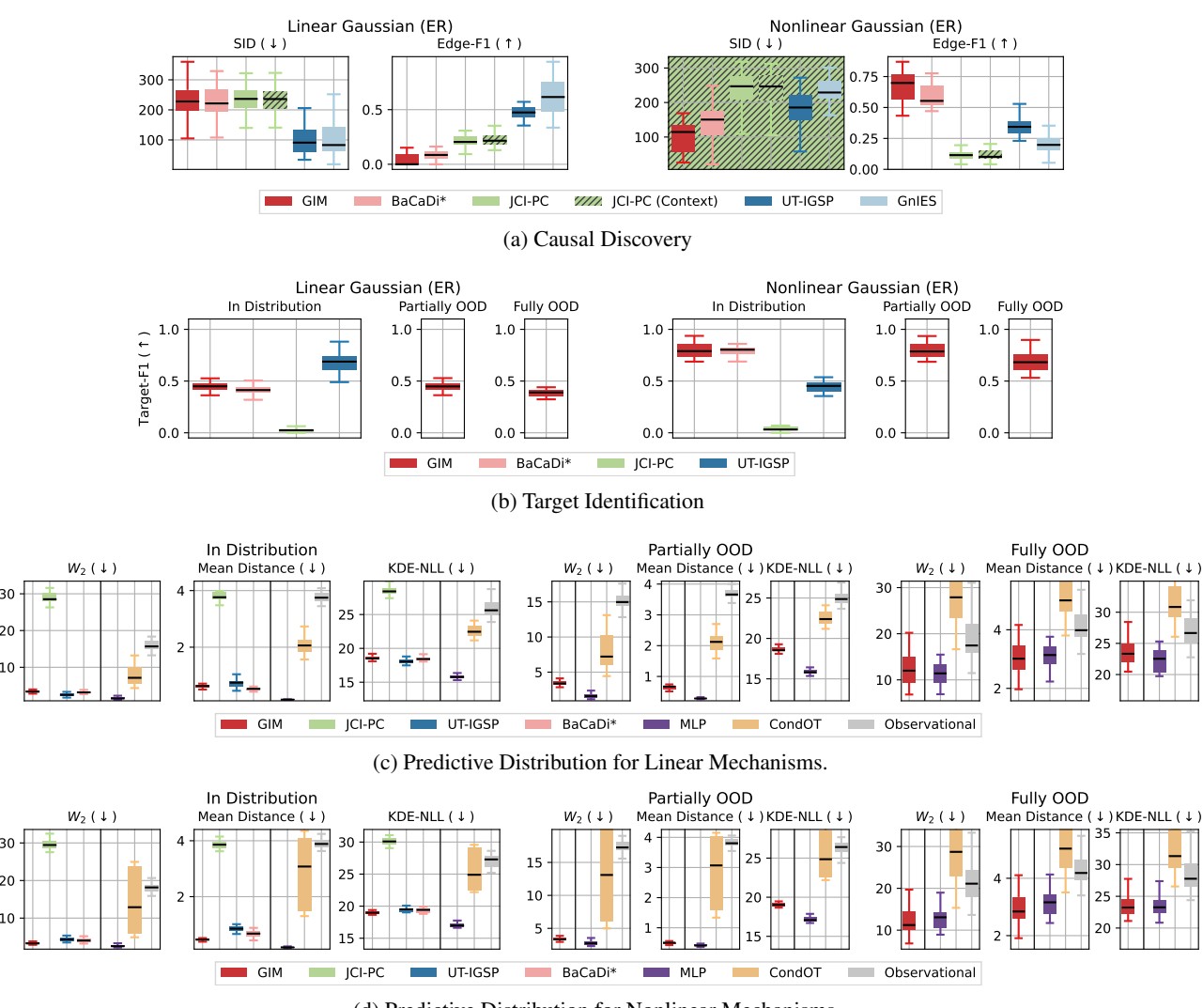

Figure 7. **Evaluating the learned causal structure, interventions and distribution shifts on Gaussian SCMs with ER graphs and shift interventions.** (a) SID and Edge-F1 scores of inferred causal graphs for linear (left) and nonlinear (right) Gaussian SCMs. (b) Target-F1 scores for in- and out-of-distribution perturbation features $\gamma$ for linear(left) and nonlinear (right) Gaussian SCMs. (c) and (d) $W_2$, Mean Distance and KDE-NLL for SCMs with linear (c) and nonlinear (d) mechanisms. We show perturbations with training dosages and targets (left, in distribution), novel dosages with seen targets (center, partially OOD), and novel targets with seen dosages (right, fully OOD). Metrics are medians over all perturbations for a given dataset. Observational baseline uses the unperturbed distribution as a prediction. Vertical line separates mechanistic and black-box approaches. Boxplots show medians and interquartile ranges (IQR). Whiskers extend to the farthest data point within $1.5 \cdot$ IQR from the boxes.

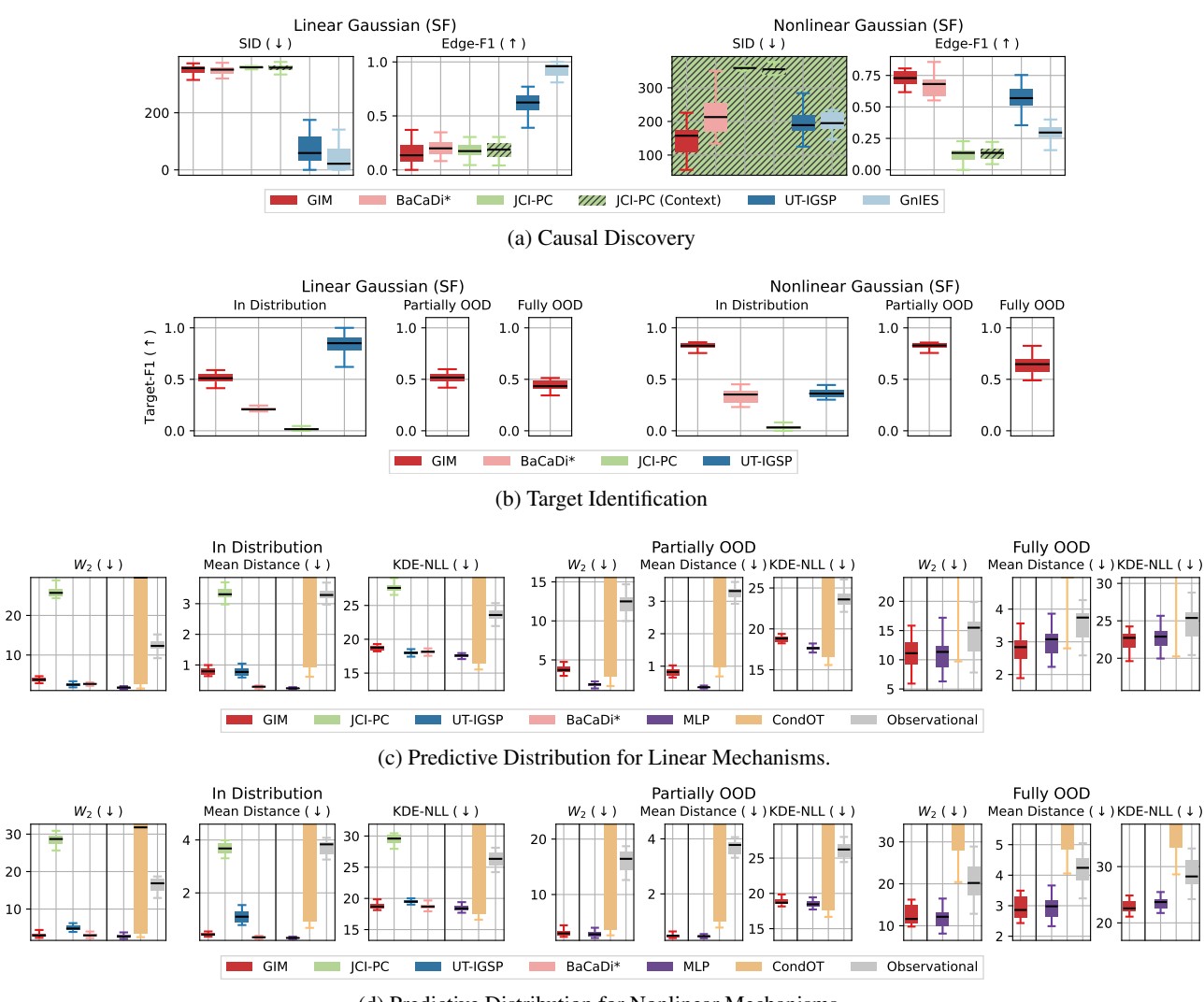

*Figure 8.* **Evaluating the learned causal structure, interventions and distribution shifts on Gaussian SCMs with SF graphs and hard interventions.** (a) SID and Edge-F1 scores of inferred causal graphs for linear (left) and nonlinear (right) Gaussian SCMs. (b) Target-F1 scores for in- and out-of-distribution perturbation features $\gamma$ for linear(left) and nonlinear (right) Gaussian SCMs. (c) and (d) $W_2$, Mean Distance and KDE-NLL for SCMs with linear (c) and nonlinear (d) mechanisms. We show perturbations with training dosages and targets (left, in distribution), novel dosages with seen targets (center, partially OOD), and novel targets with seen dosages (right, fully OOD). Metrics are medians over all perturbations for a given dataset. Observational baseline uses the unperturbed distribution as a prediction. Vertical line separates mechanistic and black-box approaches. Boxplots show medians and interquartile ranges (IQR). Whiskers extend to the farthest data point within $1.5 \cdot$ IQR from the boxes.

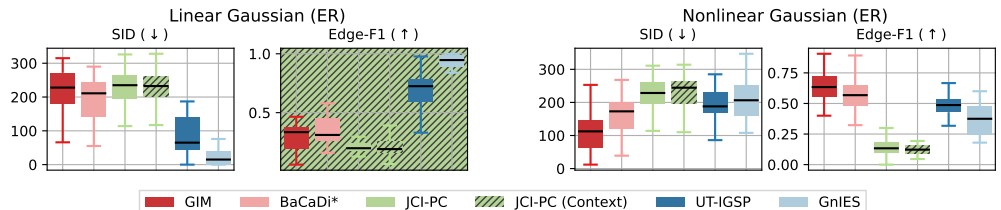

*Figure 9.* **Evaluating the learned causal structure, interventions and distribution shifts on Gaussian SCMs with SF graphs and shift interventions.** (a) SID and Edge-F1 scores of inferred causal graphs for linear (left) and nonlinear (right) Gaussian SCMs. (b) Target-F1 scores for in- and out-of-distribution perturbation features $\gamma$ for linear(left) and nonlinear (right) Gaussian SCMs. (c) and (d) $W_2$, Mean Distance and KDE-NLL for SCMs with linear (c) and nonlinear (d) mechanisms. We show perturbations with training dosages and targets (left, in distribution), novel dosages with seen targets (center, partially OOD), and novel targets with seen dosages (right, fully OOD). Metrics are medians over all perturbations for a given dataset. Observational baseline uses the unperturbed distribution as a prediction. Vertical line separates mechanistic and black-box approaches. Boxplots show medians and interquartile ranges (IQR). Whiskers extend to the farthest data point within $1.5 \cdot$ IQR from the boxes.

*Figure 10.* **Causal Discovery when Selecting Hyperparameters based on Edge-F1.** SID and Edge-F1 scores of the inferred causal graphs for linear (left) and nonlinear (right) Gaussian SCMs with hard, atomic interventions. Boxplots show medians and interquartile ranges (IQR). Whiskers extend to the farthest data point within $1.5 \cdot$ IQR from the boxes.

Varying Number of PCA Components

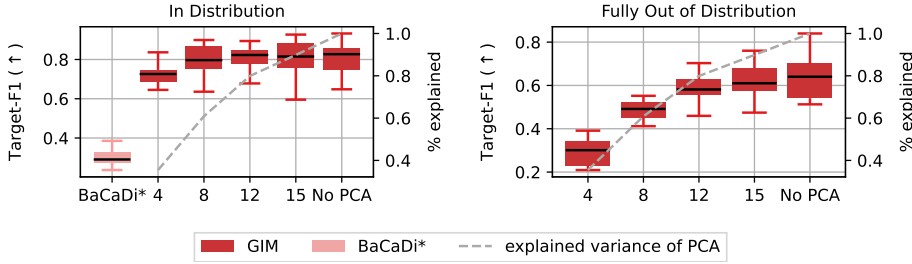

*Figure 11.* **Target Identification (Target-F1) of GIMs relative to information contained in γ.** Target-F1 scores for in-distribution (left) and fully OOD perturbations (right) in nonlinear systems with hard atomic interventions. Boxplots show medians and interquartile ranges (IQR). Whiskers extend to the farthest data point within $1.5 \cdot$ IQR from the boxes.

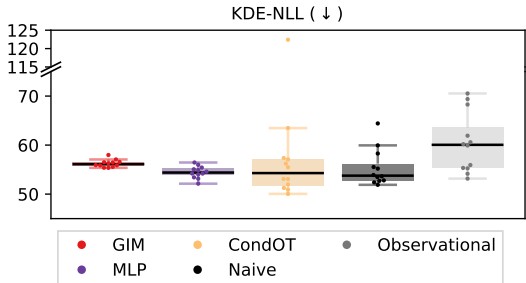

*Figure 12.* **KDE-NLL for heldout drug-dosage combinations on SciPlex3 scRNA-seq drug perturbation data** (Srivatsan et al., 2020).

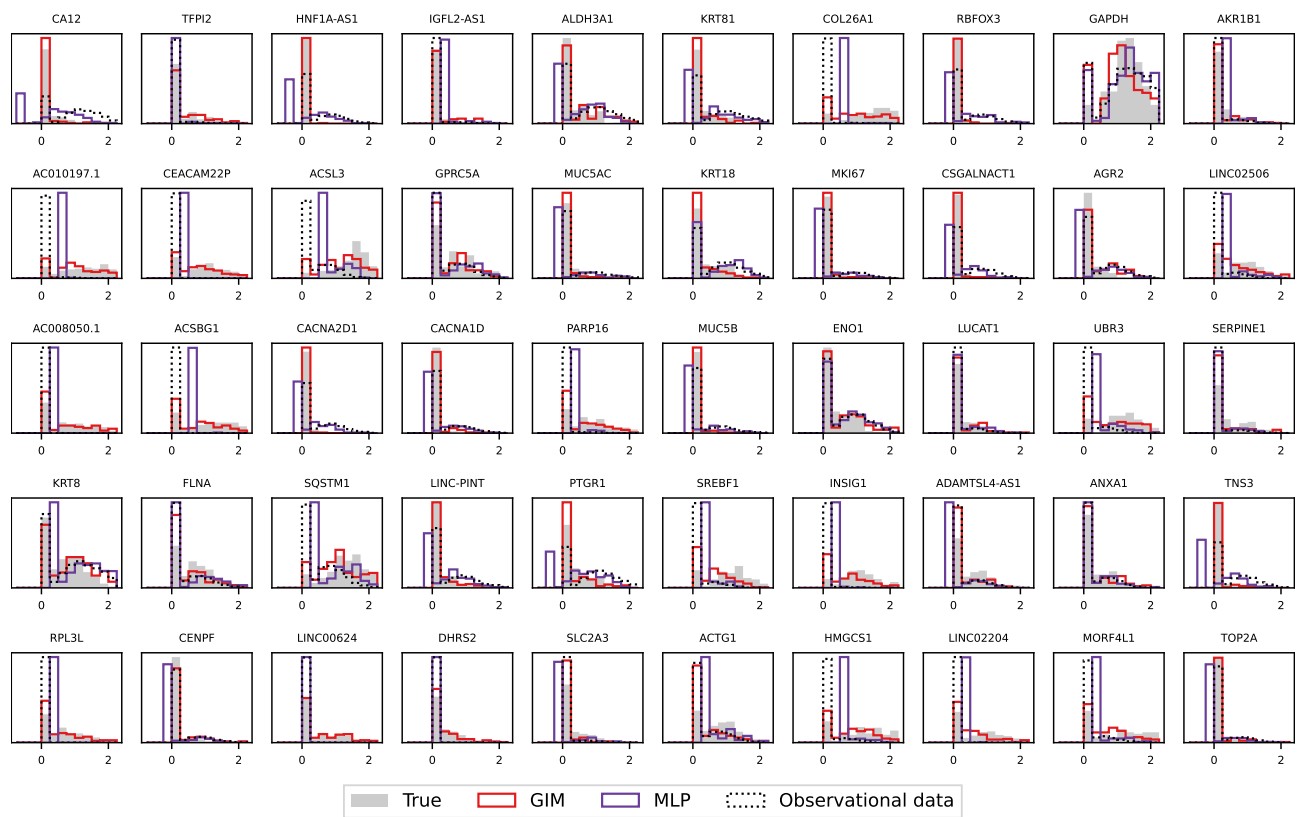

*Figure 13.* **True and predicted marginal distributions on SciPlex3 data.** 50 top-ranked marker genes for the Givinostat perturbation on A549-cells.

