# OpenReview forum: "Generative Intervention Models for Causal Perturbation Modeling"
_ICML.cc/2025/Conference — ICML 2025 poster_

### Official Review · Reviewer_rdKL · 2025-03-01

**Overall Recommendation:** 3

**Summary:**

This paper presents Generative Intervention Models (GIMs), a causal modeling framework designed to predict the effects of perturbations in complex systems with unknown underlying mechanisms. GIMs establish a mapping between perturbation features and a distribution over atomic interventions within a jointly inferred causal model. This method facilitates the prediction of distribution shifts caused by unseen perturbations. The authors assess GIMs using synthetic datasets and single-cell RNA sequencing (scRNA-seq) drug perturbation data, demonstrating that GIMs outperform traditional causal inference methods in structure and intervention target identification while maintaining predictive accuracy comparable to black-box machine learning models.

**Claims And Evidence:**

The effectiveness of the proposed method is empirically well supported.

However, it relies on black-box modeling and lacks theoretical guarantees or bounds on the correctness of its predictions. This limitation may restrict its applicability in real-world settings.

**Essential References Not Discussed:**

Overall, the paper presents key literature on causal modeling. However, since I am not familiar with generative intervention modeling, I am unable to assess whether the paper includes adequate baselines for comparison.

**Experimental Designs Or Analyses:**

The experimental design is sound. Some limitations regarding experiments are described in the evaluation section above.

**Methods And Evaluation Criteria:**

The authors conducted evaluations across various settings, including synthetic and scRNA-seq datasets. The evaluation criteria are well-chosen and provide detailed insights into the method's effectiveness.

One experimental limitation is the lack of analysis on larger-scale settings with hundreds of thousands of nodes. (Authors subsample 50 genes for scRNA-seq experiments.) It raises the question of whether the method faces computational challenges when applied at such a scale.

**Other Comments Or Suggestions:**

.

**Other Strengths And Weaknesses:**

Strengths:
- Provides a comprehensive overview of existing literature.

Weaknesses:
- The technical novelty is limited, as it adopts an existing Bayesian structure learning framework, e.g., Brouillard et al., "Differentiable Causal Discovery from Interventional Data," NeurIPS 2020.

**Questions For Authors:**

- What is the computational complexity of the proposed method?
- Is the method applicable to complete gene datasets containing thousands of genes?

**Relation To Broader Scientific Literature:**

The key contributions of this paper are applicable to a broad range of scientific domains, particularly in the biological field.

**Theoretical Claims:**

This paper does not contain any theoretical claims for their method.

---

> ### Author Rebuttal · Authors · 2025-04-01
>
> Thank you for your detailed comments and feedback. We appreciate that you recognize the strong empirical results of the work and the potential impact in scientific domains such as biology.  You raised several important points that we address in detail below. Please let us know if you have any remaining concerns or questions, and we would be happy to clarify further.
>
> > it relies on black-box modeling and lacks theoretical guarantees or bounds on the correctness of its predictions.
>
> While our approach leverages neural networks, it is specifically designed *not* to be a purely black-box predictor. GIMs are built on a causal modeling framework with interpretable, mechanistic components: a structural causal model $\mathcal{M}$ and interventions $\mathcal{I}$, which model how perturbations manifest in $\mathcal{M}$. This allows us to not only predict the effect of a perturbation, but also understand qualitatively how it alters the system’s underlying causal structure and mechanisms.
>
> While a full theoretical analysis is beyond the current scope, existing identifiability results from causal discovery with unknown interventions (e.g., Brouillard et al., 2020) apply to our setting under standard assumptions. We discuss this in detail in our response to Reviewer 8P9S.
>
> > technical novelty is limited, as it adopts an existing Bayesian structure learning framework, e.g., Brouillard et al.
>
> GIMs introduce a new modeling framework that uses causal models to predict the effects of previously unseen, general perturbations, a task that standard causal discovery methods are not designed to handle. While we build on existing techniques for inferring the causal model $\mathcal{M}$ (e.g., prior design and gradient-based optimization), we introduce an explicit  model of how perturbation features induce interventions in the causal model. Specifically, GIMs learn a shared generative mechanism that maps observable perturbation features $\gamma$ to distributions over latent interventions, enabling generalization to entirely new perturbations. In contrast, methods such as Brouillard et al. (2020) identify the unknown intervention targets for each training perturbation without incorporating their features, typically with the goal of recovering the causal graph. This limits their use to scenarios where the goal is recovering the causal graph and predicting effects for known perturbations, making them unsuitable for predicting the effects of new, unseen perturbations.
>
> >  What is the computational complexity of the proposed method?
>
> The computational complexity of GIMs is comparable to prior causal modeling approaches such as Hägele et al. (2023) and Brouillard et al. (2020) since we use similar causal model classes. The only additional overhead introduced by our framework is the forward pass through the GIM MLP, which does not affect the asymptotic complexity. In terms of wall-clock time, for a nonlinear system with 20 variables (as in Figure 2B), training a full GIM takes on average 20.71 ± 0.33 minutes on a GPU, averaged over 20 random datasets. For reference, BaCaDi* takes 11.56 ± 0.11 minutes under the same conditions.
>
> Formally, the computational complexity of learning GIMs is:
> $$ \mathcal{O}\left( T\cdot \left[ d^2 \cdot \left(p_z + H_{\mathcal{M}} + n_{\text{MC}} n_{\text{power}}\right) + n_{\text{MC}} \cdot n_{\text{total}} \cdot d \cdot L_{\mathcal{M}} \cdot H_{\mathcal{M}}^2 + K \cdot \left( L_{\text{GIM}} \cdot H_{\text{GIM}}^2 + H_{\text{GIM}} \cdot (p + d) + n_{\text{MC}} \cdot d \right) \right]
> \right), $$
> where:
> - $T$: number of training steps,
> - $d$: number of variables in $\mathbf{x}$,
> - $p_z$: rank-parameter of $\mathbf{Z}$,
> - $n_{MC}$: number of MC samples,
> - $n_{\text{power}}$: number of power iterations used in NO-BEARS acyclicity,
> - $n_{\text{total}}$: total number of samples across environments,
> - $L_{GIM}, L_{\mathcal{M}}$: depth of GIM MLPs and causal mechanisms, respectively,
> - $H_{GIM}, H_{\mathcal{M}}$: width of GIM MLPs and causal mechanisms, respectively,
> - $p$: dimension of $\gamma$,
> - $K$: number of perturbation environments.
>
> We added this to the supplementary material of our manuscript.
>
> > method applicable to complete gene datasets containing thousands of genes?
>
> Like comparable causal modeling approaches, GIMs do not scale easily to very large systems. Since our focus is on introducing a new modeling framework for perturbation effects, scalability is not a primary concern – in particular, because our experiments on scRNA-seq data demonstrate that GIMs can achieve strong predictive performance in real-world settings, even when applied to a subset of genes. That said, we agree that scalability is important and may be improved in future work. Approaches developed for scaling causal models, for example modeling the graph $\mathbf{G}$ as a low-rank factor graph (Lopez et al., 2022), can be naturally integrated into our framework by modifying the causal model $\mathcal{M}$ accordingly.

---

> > ### Comment · Reviewer_rdKL · 2025-04-04
> >
> > I appreciate the detailed rebuttal! The detailed analysis of computational complexity and scalability is informative. Based on the rebuttal, I have updated my score to weak accept.

---

### Official Review · Reviewer_feSa · 2025-03-08

**Overall Recommendation:** 3

**Summary:**

This paper studies the problem of predicting the unseen perturbation effect in a causal model. The authors propose the Generative Intervention Model (GIM) framework to learn the relationship between perturbation and distribution shift in a causal model. It is claimed that the GIM can predict perturbation features that do not show up in the training data. Detailed experiments on synthetic data and scRNA-seq are done to showcase the good performance of the GIM methods compared to other causal inference methods.

**Claims And Evidence:**

Claims made in the submission supported by clear and convincing evidence.

**Essential References Not Discussed:**

Relevant works are discussed.

**Experimental Designs Or Analyses:**

The experiments are well-designed. Implementation details are provided.

**Methods And Evaluation Criteria:**

The paper uses synthetic data and scRNA-seq datasets in the experiments. The entropy-regularized Wasserstein distance between predicted and ground-truth distribution and the Euclidean distance between the mean are used as metric.

**Other Comments Or Suggestions:**

It would be good if the authors could add a concrete real-world example in the third section to illustrate the terminologies. It took me a while to understand all the notations. In particular, explanations of $\gamma$ and how it may influence intervention are needed. Besides, i am still a bit confused about what is in the dataset $\mathcal{D}$. Is intervention $I$ in the dataset?

**Other Strengths And Weaknesses:**

Strengths: The proposed GIMs leverage perturbation features to predict distribution shifts in unseen conditions, which enables to model to generalize beyond seen distribution. Experimental results show that GIMs outperform classical approaches in identifying causal structures and intervention targets in nonlinear systems.

Weaknesses: Jointly estimating both the causal model and the generative intervention mapping seems to be a complex optimization problem, which may be computationally expansive.

**Questions For Authors:**

Thanks for the great work! It will be great if the author can clarify the following questions.

1. As I mentioned above, can you explain the formulation more clearly? In particular, what are known and what are unknown?

2. The authors also mention in the paper that [1] uses a similar Bayesian method for structure learning. Can you explain the similarity and difference between your work and [1]?

3. Can the author explain a bit about how acyclicity is ensured when sampling from the prior?

I also wonder if the authors can provide the code in a later version.

[1] Lorch, Lars, et al. "Dibs: Differentiable bayesian structure learning." Advances in Neural Information Processing Systems 34 (2021): 24111-24123.

**Relation To Broader Scientific Literature:**

Several related works are mentioned in the Introduction part

**Theoretical Claims:**

I did not check the derivations in Appendix B carefully, but they seem correct.

---

> ### Author Rebuttal · Authors · 2025-04-01
>
> Thank you for your detailed comments and feedback. We are glad to hear that you find our work to be supported by clear and convincing evidence, as well as our experiments to be well designed. You raised several important points that we address in detail below.
> > Jointly estimating both the causal model and the generative intervention mapping seems to be a complex optimization problem, which may be computationally expansive.
>
> The computational complexity indeed increases compared to settings, where only the causal model and intervention targets are directly inferred (e.g. BaCaDi (Hägele et al., 2023)). However, in practice, we find the overhead to be manageable. For example, in the 20-node nonlinear setting (as used for Figure 2B) and training on a GPU, BaCaDi* takes on average 11.56 ± 0.11 minutes, while GIM takes 20.71 ± 0.33 minutes. Both results are averaged over 20 random datasets.
> > add a concrete real-world example [....] & explain the formulation more clearly? In particular, what are known and what are unknown?
>
> A concrete real-world example is drug perturbation experiments, such as those on scRNA-seq data that we present in Section 6, Figure 5. We perform $K$ experiments, where in each one a drug is applied to cells and we measure their responses - typically via its gene expression profiles $\mathbf{x}$. In experiment $k$, we observe the response of $n_k$ cells, yielding $n_k$ samples of $\mathbf{x}$, denoted as $\mathbf{X}^{(k)}$. The perturbation features $\gamma^{(k)}$ capture observable drug properties, such as identity, molecular characteristics, or dosage. The full dataset $\mathcal{D}$ consists of $K$ pairs of drug features and responses: $(\gamma^{(k)}, \mathbf{X}^{(k)})$ for $k = 1, \dots, K$. Our goal is to predict the response of a new drug $\gamma^*$, i.e. to predict the gene expression under this new drug.
>
> The *latent* causal model $\mathcal{M}$ defines the data-generating process of $\mathbf{x}$. A perturbation $\gamma^{(k)}$ influences this system through an *unobserved* intervention $\mathcal{I}^{(k)}$. The distribution over these interventions, conditioned on $\gamma^{(k)}$, is parameterized by *latent* parameters $\phi$, which are shared across all experiments and enable generalization to new perturbations.
>
> In summary, we observe $K$ pairs of $(\gamma^{(k)}, \mathbf{X}^{(k)})$. The causal model $\mathcal{M}$, the interventions $\mathcal{I}^{(k)}$, and the parameters $\phi$ are unobserved and jointly inferred. We added this clarification in line 134.
> > similarity and difference between your work and [Lorch et al. (2021), Hägele et al. (2023)].
>
> GIMs introduce a new modeling framework that uses causal models to predict the effects of previously unseen, general perturbations, a setting that causal discovery methods like Lorch et al. (2021) and Hägele et al. (2023), are not designed to handle. While we adopt a similar prior design for the causal model $\mathcal{M}$ and use gradient-based inference, we introduce a fundamentally different approach to modeling perturbations.
> Lorch et al. assume that the intervention $\mathcal{I}$ is observed for each perturbation; unlike GIMs, their method cannot be applied when interventions are unknown. Hägele et al. infer interventions for each training perturbation, but treat each perturbation independently, without leveraging any features. As a result, their model cannot generalize to novel perturbations beyond those seen during training. In contrast, we propose to learn how interventions are induced by the observable perturbation features $\gamma$. Specifically, GIMs learn a shared generative mechanism that maps observable perturbation features $\gamma$ to a distribution over latent interventions, enabling generalization to entirely new, unseen perturbations. Thus, while prior approaches focus on recovering the causal graph or inferring training interventions, GIMs are designed for the task of predicting the effects of new perturbations.
>
> > how acyclicity is ensured when sampling from the prior?
>
> We enforce acyclicity via constrained optimization using the augmented Lagrangian approach, as introduced in Zheng et al. (2018) and also used in related works, such as Brouillard et al. (2020). While this encourages acyclic graphs during training, there is no hard guarantee that the estimated graph is cycle-free. In practice, we randomly break cycles if they occur.
>
> > provide the code in a later version
>
> We will provide the full code in the camera-ready version.

---

> > ### Comment · Reviewer_feSa · 2025-04-04
> >
> > Thanks for the detailed explanation. It is much clearer to me now. I suggest briefly mentioning the drug example in section 3 to help the reader understand the setting better.

---

### Official Review · Reviewer_8P9S · 2025-03-15

**Overall Recommendation:** 3

**Summary:**

The authors considered the problem of predicting the impact of interventions with applications in gene perturbation prediction. In particular, in some applications, when an intervention is performed, it is unknown which variables are intervened on. However, some features of the intervention might be known. The authors motivated this by an example in genomics, where features of a drug might be known but it is not clear what is the causal effect of a drug on regulatory pathways in a cell. The authors considered a causal modeling approach so that the outputs are easily interpretable. More specifically, they trained a generative model to map the features of a perturbation to a distribution over atomic interventions in the system. The experimental results showed that for the task of causal discovery or intervention target identification, the proposed method has better performance compared to previous work. Moreover, for predicting the effect of perturbation in out-of-distribution settings, the proposed method achieved similar performance compared to methods that do not use causal modeling in their architectures.

### Update after rebuttal
I increased my score to 3 after the discussion on the theoretical guarantees. However, as I mentioned before, the current paper is somehow a natural extension of Lorch et al. (2021) and Hägele et al. (2023), particularly in scenarios where intervention targets are not given or where features are not used to locate intervention targets. Therefore, I am not sure about the technical novelty in the current work and not giving a higher score.

**Claims And Evidence:**

The work is more experimental although it also provides proof for the derivations of some of the equations in the appendix. It seems that the authors made a good comparison with previous work.

**Essential References Not Discussed:**

As far as I checked the related work, the authors cited the most relevant previous work.

**Experimental Designs Or Analyses:**

I did not check the codes but did read the experiment section. Based on the text, it seems that the experimental results are sound and the authors also provided some explanations for the plots.

**Methods And Evaluation Criteria:**

They considered several metrics for comparison such as SID and Edge-F1 for causal discovery or $W_2$ and mean distance for the task of predicting the impact of perturbations. Moreover, they considered both synthetic and real datasets in their evaluations.

**Other Comments Or Suggestions:**

It would be nice to study whether some theoretical guarantees can be added to the work in some specific settings such as linear settings. Moreover, it is good to characterize how the contained information in perturbation features $\gamma$ affects perturbation prediction. For this purpose, some simple settings such as bivariate models might be considered to facilitate the analysis.

**Other Strengths And Weaknesses:**

Strengths:
- The authors proposed a method to train a causal generative model that is more interpretable (for instance, checking which variables are intervened) compared to previous work, especially the unstructured models.

- They performed comprehensive experiments in various settings, showing that the proposed method is better or on par with previous works.

Weaknesses:
- The main weakness is that there is no theoretical guarantee in the work. That being said, I should emphasize that this comment is also applied to some of the previous work.

- In deriving the objective function of the optimization problem, several design choices/approximations are considered. It is not clear what are the impacts of such approximations and why it is fine to consider them.

**Questions For Authors:**

- In line 184, what is $p$ in the definition of the domain of $Z$? What are $z_{0i}$ and $z_{1j}$ in line 190? Please explain in more detail the modeling in line 190?
- In eq. (11), please explain more about the design choice for target sparsity.
- In the experiments, it was observed that the proposed method misses an intervention on a variable but predicts an intervention on its parent or ancestor. It is good to explain this observed phenomenon.
- It seems that parts of the proposed method were borrowed from previous work such as Lorch et al. (2021) or Hagele et al. (2023). It is good to mention that other than the generative process of $I$ given $\gamma$, what are the main contributions compared to previous work?
- I think the performance of the method highly depends on $M$. It is good to study experimentally the effect of $M$ on the performance. I am geussing $M$ should be very large in order to have reasonable performance.

**Relation To Broader Scientific Literature:**

One of the main applications of causal inference/discovery is in biology such as learning causal structures in gene regulatory networks or predicting the results of gene perturbations. This work is aligned with this line of research and aims to predict the effect of intervention when it is unknown which variables are intervened by giving the treatment.

**Theoretical Claims:**

The work is experimental and there is no theoretical claim in the paper.

---

> ### Author Rebuttal · Authors · 2025-04-01
>
> Thank you for your detailed feedback. We are glad to hear that you find our problem setting relevant, the experiments comprehensive, and the interpretability of our approach valuable. You raised several important points that we address below. Please let us know if you have any remaining concerns or questions, and we would be happy to clarify further.
> > no theoretical guarantee [...] I should emphasize that this comment is also applied to some of the previous work.
>
> We propose a new modeling framework that generalizes to unseen perturbations and demonstrates strong empirical performance on simulated and real-world data. While a full theoretical analysis is beyond the scope of this paper, we agree that identification guarantees are important. Below, we describe how existing results apply directly to GIMs, which allow contextualizing our results. We added these explanations to line 180.
> Under standard assumptions, the MAP optimizers in GIMs identify the true intervention targets of the training perturbations, $\mathbb{I}$, and a graph $\mathbb{I}$-Markov equivalent to the true one. To establish this, we extend Theorem 2 from Brouillard et al. (2022). They prove that, under standard assumptions and with sufficiently small regularization, the score $\mathcal{S}(\mathbf{G},\mathbb{I})$ is maximized by the true intervention targets and an $\mathbb{I}$-equivalent graph. Assuming the GIM prior distributions have global support, we recover the same maximizers in the large-sample limit, because the posterior is dominated by the likelihood: $\arg\max_{\mathbf{G}, \mathbb{I}} \mathcal{S}(\mathbf{G},\mathbb{I})=\arg\max_{\mathbf{G},\mathbb{I}}\sup_{θ}\log p(θ, \mathbf{G},\mathbb{I}\mid\mathcal{D})$. Assuming that there exists a mapping from features to the true targets and it can be expressed by $\phi$, we have: $\max_{\mathbf{G}, \mathbb{I}}\sup_{θ}\log p(θ,\mathbf{G},\mathbb{I}\mid\mathcal{D})=\max_{\mathbf{G},\phi} \sup_{θ}\log p(θ,\mathbf{G},\phi\mid\mathcal{D})$. The MAP optimizers recover $\mathbb{I}$ and a graph $\mathbb{I}$-Markov equivalent to the true one. Identifiability of interventions for unseen perturbations depends on the informativeness of $γ$, and we empirically show the impact on predictive performance in Figure 4.
> > design choices/approximations are considered [...] why it is fine to consider them.
> [....] explain more about the design choice for target sparsity
>
> We approximate expectations in the prior and likelihood using Monte Carlo sampling, and we use the Gumbel-Softmax relaxation for differentiable sampling of discrete variables. The MC estimators are consistent, while the Gumbel-Softmax introduces a bias that vanishes as the sigmoid temperature $τ \to 0$. In practice, we fix $τ$. These approximations are widely used in the structure learning literature.
> Our prior design follows standard causal modeling assumptions, including sparsity and acyclicity for the graph and Gaussian priors over parameters for regularization. For the intervention targets, we use an L1-based sparsity-inducing prior, reflecting the assumption that typically only a few variables are affected per perturbation. This aligns with the sparse mechanism shift hypothesis, which suggests that interventions tend to induce localized changes (Schölkopf, 2022). All priors have full support; thus, in the large-sample limit, their impact diminishes.
> > what is $p$ [...]? What are $z_{0i}$ and $z_{1j}$ [...]?
>
> $p$ is a parameter controlling the rank of the matrices of $p(\mathbf{G}\mid Z)$. For $p\geq d$, this distribution can represent any adjacency matrix without self-loops. In all experiments, we set $p=d$. The tensor $Z$ consists of two $d \times p$ matrices; $z_{0i}$ and $z_{1j}$ refer to the $i$-th row of the first and the $j$-th row of the second matrix, respectively. We added this explanation in line 193.
> > [In Figure 2C, the] proposed method misses an intervention on a variable but predicts an intervention on its parent
>
> We show this example to discuss how GIMs behave in practice and how they may infer consistent but nevertheless incorrect interventions in certain cases. Here, the true intervention targets $X_i$, and in the true graph, $X_j$ is a child of $X_i$. The intervention shifts the distribution of $X_i$, which affects $X_j$ via $p(X_j \mid X_i)$. In the estimated graph, the edge is reversed and the model instead attributes the shift to an intervention on $X_j$. While this is incorrect under the true graph, it is consistent with the predicted graph and still describes the observed distribution.
> > main contributions compared to previous work
>
> Please see our response to reviewer feSa.
> > method highly depends on $M$
>
> The predictive performance can be affected by the number of MC samples $M$.  In all experiments, we used $M=128$. We additionally tested smaller $M$ in the nonlinear setting (Figure 3) and found no significant effect on accuracy:
> | |M=2|M=32|M=128|
> |---|---|---|---|
> |$W_2$|17.78 ± 5.35|17.69 ± 4.99|18.13 ± 4.33|

---

> > ### Comment · Reviewer_8P9S · 2025-04-02
> >
> > Thank you for the responses. I suggest adding an explanation of the theoretical guarantees, as well as a discussion on the impact of M on the accuracy, in the revised version. I have adjusted my score accordingly. However, I still think that the current paper is somehow a natural extension of Lorch et al. (2021) and Hägele et al. (2023), particularly in scenarios where intervention targets are not given or where features are not used to locate intervention targets.

---

### Official Review · Reviewer_jgn6 · 2025-03-16

**Overall Recommendation:** 3

**Summary:**

This paper studies the problem of causal perturbation modeling to recover the causal structure and intervention targets given perturbation features from several interventional environments. The use-case in this paper is the gene perturbations in the biology domain. The authors propose generative intervention model (GIM) which learns to map perturbation features to atomic intervention vectors (1-sparse) in a jointly-estimated causal model. This modeling enables the ability to infer the effect of unseen perturbation features during inference. Thus, this approach is robust to distribution shifts in the feature perturbation space. Experiments are conducted on synthetic and scRNA-seq drug perturbation data to show the effectiveness of the proposed method.

## Update After Rebuttal
I thank the authors for the detailed response to my questions and concerns. I believe this is an interesting work that relaxes the major assumption of having intervention targets available in causal representation learning. Furthermore, I believe the application in biology is quite interesting. Therefore, I lean towards **acceptance** of this paper.

**Claims And Evidence:**

Yes

**Essential References Not Discussed:**

N/A

**Experimental Designs Or Analyses:**

Yes, I checked the metrics used, including the Structural Intervention Distance (SID) and Wasserstein distance for evaluation. Furthermore, I checked the soundness of the data-generation with respect to the structural causal models and the empirical setup.

**Methods And Evaluation Criteria:**

Yes

**Other Comments Or Suggestions:**

N/A

**Other Strengths And Weaknesses:**

## Strengths

- Overall, the paper is written well with clear intuitions about the problem of causal intervention modeling. Furthermore, this paper seems to make weaker assumptions about access to interventional data, specifically that one would only have access to the perturbation features and not the entirety of the interventional data for all environments, which is a more realistic scenario in practice.
- The application in single-cell gene perturbations is certainly an interesting use-case with great potential impact, especially for reasoning about hypothetical perturbations outside the support of the training distribution.
- The partial OOD (novel dosages with seen targets) and fully OOD (novel targets with seen dosages) is a robust way to evaluate the effects of unseen perturbations. Specifically, the fully OOD setting shows promising results compared to other methods.

## Weaknesses

- The overall objective can use more clarity. The idea seems to be to parameterize a network $\phi$ to infer the intervention target given the perturbation vector. To learn a joint causal model, one objective is to learn a latent variable $z$ to model the causal graph $G$ probabilistically. The overall goal is to maximize a log-likelihood with respect to the causal model and intervention prediction network over the dataset.

**Questions For Authors:**

- What is the main difference between this approach for extrapolating to unseen perturbations and the causal representation learning method proposed by Zhang et al? In this method, the authors show identifiability guarantees from soft interventions and recover the causal factors and their relationships up to an equivalence class given sufficient interventional data.
- Intuitively, what is the meaning of the perturbation vector? Is this simply the perturbation that generates K different interventional environments? Is it a value vector that takes a different value across environments?
- From my understanding, this setting does not explicitly require access to interventional data. Rather, one requires only a perturbation vector. Is this interpretation correct?
- The authors claim that multiple interventions can also be supported by simply inferring each one separately from the perturbation features. However, how does this work in practice?

Zhang et al. Identifiability Guarantees for Causal Disentanglement from Soft Interventions. NeurIPS 2023.

**Relation To Broader Scientific Literature:**

Overall, this paper tackles an important problem in causal generative modeling. Specifically, for domains such as biology, predicting the effects of unseen interventions in critical for scientific discovery. Along a line of work that focuses on causal models for biological applications, such as intervention design and causal representation learning, this work explores a different approach to interventional inference.

**Theoretical Claims:**

N/A

---

> ### Author Rebuttal · Authors · 2025-04-01
>
> Thank you for your detailed comments and feedback. We are glad to hear that you find our approach addresses an important problem in causal generative modeling with an interesting use-case with great potential. You raised several important points that we address in detail below.
>
> > overall objective can use more clarity.
>
> We revised the manuscript in lines 95ff, 133ff, 154ff to further clarify.
>
> > What is the main difference between this approach for extrapolating to unseen perturbations and the causal representation learning method proposed by Zhang et al?
>
> Thank you for pointing us to the work by Zhang et al. (2023). Their setting differs from ours: their causal model is defined over latent (unobserved) variables, whereas GIM models interventions over observed variables (e.g., gene expression). This makes GIM’s predictions more interpretable, as it directly models causal effects between semantically meaningful variables. While it is unknown in both approaches how the perturbation modifies the underlying causal model, the two methods tackle this challenge in fundamentally different ways. Zhang et al. (2023) infer interventions for each perturbation in the training data, independent of any features. In contrast, we propose to learn how interventions are induced by the observable perturbation features $\gamma$. Since Zhang et al. (2023) do not incorporate such features, their model *cannot generalize to novel perturbations* beyond those seen during training, similar to Hägele et al. (2023). Zhang et al.’s approach is limited to combinations of previously observed interventions, while GIMs enable predictions for entirely unseen new perturbations by conditioning explicitly on $\gamma$.
> We also note that, because GIMs model the causal variables directly, known identifiability results can be extended to GIMs (see our response to Reviewer 8P9S). Overall, we view the two approaches as complementary, exploring different but important aspects of causal modeling with perturbations.
> > meaning of the perturbation vector
>
>
> $\gamma$ encodes any observable information about the perturbation applied in a given environment, such as drug properties and dosage in a drug perturbation experiment. For example, in the scRNA-seq setting, we define it as a one-hot encoding of the drug along with its dosage as the perturbation features. The perturbation vectors $\gamma$ indeed differ across experimental environments. For each environment (i.e., context or experiment), we have one perturbation vector $\gamma \in \mathbb{R}^p$, and a corresponding collection of samples of the observed variables $\mathbf{x}$, such as gene expression levels. In line 78, we adjusted the manuscript to clarify the meaning of $\gamma$ and that it differs across environments.
>
> > this setting does not explicitly require access to interventional data. Rather, one requires only a perturbation vector. Is this interpretation correct?
>
> Yes, your interpretation is correct. GIMs do not require access to interventional data in the sense of having knowledge of the atomic intervention $\mathcal{I}$ corresponding to a perturbation — that is, which variables in the causal model were directly manipulated and how. During training, we assume access to $K$ different perturbation experiments, where for each experimental context we observe perturbation features $\gamma$, along with samples of observed variables $\mathbf{x}$ collected under that condition. At test time, GIMs only require a perturbation vector $\gamma^*$ to predict the system’s response. This setup reflects many realistic biological settings, where intervention targets are unknown but perturbation metadata is available.
>
> > multiple interventions can also be supported by simply inferring each one separately from the perturbation features. However, how does this work in practice?
>
> Our approach handles multiple perturbations (i.e., environments) by inferring a distribution over interventions $\mathcal{I}$ — that is, intervention targets and parameters — for each perturbation individually based on its perturbation features $\gamma$. In practice, the generative intervention model consists of two neural networks, $g_\phi$​ and $h_\phi$​, which map $\gamma$ to the parameters of this distribution: $g_\phi$​ outputs Bernoulli probabilities over targets, and $h_\phi$​ outputs the mean and variance of a Gaussian over intervention parameters. Crucially, the parameters $\phi$ are shared across all perturbation environments. Given a different $\gamma^{(k)}$, the same model predicts a distinct intervention distribution for that specific perturbation. Combined with the causal model $\mathcal{M}$, this yields a full predictive distribution over the system variables $\mathbf{x}$.
> In short, multiple perturbations are handled by applying the shared generative model to each perturbation feature vector $\gamma$, enabling per-environment inference in a unified way.

---

### Decision · Program_Chairs · 2025-05-01

**Decision:**

Accept (poster)

**Comment:**

This paper investigates causal perturbation modeling to recover the causal structure and intervention targets, given perturbation features from several interventional environments. In particular, the authors proposed a generative intervention model (GIM) that learns to map these perturbation features to distributions over atomic interventions in a jointly-estimated causal mode. In addition, the authors applied the proposed method to gene perturbation prediction and obtained promising results. All the four reviewers agreed that the paper is well written, and the effectiveness of the proposed method is well supported by empirical results. Reviewers raised some concerns about assumptions, technical details, computational complexity, novelty, etc. Many of them have been addressed by the detailed responses from authors, but some reviewers still concern about the novelty of this work, compared to existing work such as Lorch et al. (2021) and Hägele et al. (2023).